# Beyond Hearing: Learning Task-Agnostic ExG Representations from Earphones via Physiology-Informed Tokenization

**Hyungjun Yoon**[1]* **Seungjoo Lee**[2]* **Yu Yvonne Wu**[3]* **Xiaomeng Chen**[4]* **Taiting Lu**[5]
**Freddy Yifei Liu**[2] **Taeckyung Lee**[1] **Hyeongheon Cha**[1] **Haochen Zhao**[6] **Gaoteng Zhao**[7]
**Dongyao Chen**[4] **Cecilia Mascolo**[3] **Sung-Ju Lee**[1] **Lili Qiu**[8,9]

[1]KAIST   [2]Carnegie Mellon University   [3]University of Cambridge
[4]Shanghai Jiao Tong University   [5]Pennsylvania State University   [6]UCLA
[7]Northwest University   [8]University of Texas at Austin   [9]Microsoft Research

## Abstract

Electrophysiological (ExG) signals offer valuable insights into human physiology, yet building foundation models that generalize across everyday tasks remains challenging due to two key limitations: (i) insufficient data diversity, as most ExG recordings are collected in controlled labs with bulky, expensive devices; and (ii) task-specific model designs that require tailored processing (*i.e.*, targeted frequency filters) and architectures, which limit generalization across tasks. To address these challenges, we introduce an approach for scalable, task-agnostic ExG monitoring in the wild. We collected 50 hours of unobtrusive free-living ExG data with an earphone-based hardware prototype to narrow the data diversity gap. At the core of our approach is *Physiology-informed Multi-band Tokenization (PiMT)*, which decomposes ExG signals into 12 physiology-informed tokens, followed by a reconstruction task to learn robust representations. This enables adaptive feature recognition across the full frequency spectrum while capturing task-relevant information. Experiments on our new *DailySense* dataset—the first to enable ExG-based analysis across five human senses—together with four public ExG benchmarks, demonstrate that *PiMT* consistently outperforms state-of-the-art methods across diverse tasks.

## 1 Introduction

Electrophysiological (ExG) signals, including electroencephalography (EEG), electromyography (EMG), electrooculography (EOG), and electrocardiography (ECG), provide critical insights into neural, muscular, ocular, and cardiovascular activities. They enable a wide range of physiological applications, from gaze tracking (Merino et al., 2010) and emotion recognition (Gkintoni et al., 2025) to sleep staging (Nguyen et al., 2016) and seizure detection (JW et al., 2016). Recent advances in deep learning have improved ExG analysis by developing data-driven training approaches (Song et al., 2022; Jiang et al., 2024) that capture complex temporal and spectral patterns for various physiological tasks. Building on this, foundation models, which have demonstrated remarkable success across domains by leveraging large-scale data to learn general-purpose representations (Narayanswamy et al., 2025), offer a promising opportunity for advancing everyday ExG analysis.

However, ExG foundation models remain underexplored due to two limitations: (i) insufficient dataset diversity and (ii) task-specific model design. First, ExG datasets are typically collected in controlled environments (Zheng & Lu, 2015; Katsigiannis & Ramzan, 2018; Wang et al., 2023) using bulky, expensive devices (*e.g.*, EEG headsets (Duvinage et al., 2013)). This setup restricts both scale

---

*Equal contribution. The work is done during internship at Microsoft Research. Correspondence to: hyungjun.yoon@kaist.ac.kr, seungjoolee@cmu.edu, yw573@cam.ac.uk, sjtu_chenxm@sjtu.edu.cn

and diversity across tasks, leaving free-living ExG data largely untapped. Second, existing ExG models are highly task-specific, relying on tailored processing pipelines, *i.e.*, architectures optimized for a fixed frequency band, which limits their generalization. For example, gaze tracking methods are designed to capture low-frequency bands (0.1–15 Hz) (Merino et al., 2010), whereas emotion recognition relies on higher EEG bands (8–30 Hz) (Gkintoni et al., 2025). As a result, a model trained for gaze tracking cannot be directly applied to emotion recognition, highlighting the lack of transferability across tasks.

To address the first challenge, we collected free-living ExG data in unobtrusive settings, constructing the *DailySense* dataset. For this, we prototyped *NeuroBuds*, an earphone-based ExG sensing device. Unlike traditional bulky systems, NeuroBuds is lightweight, low-cost, and portable while still capturing rich physiological signals: near-ear EEG, EMG from facial muscles, and EOG from eye movements. This design enables long-term data collection, overcoming the constraints of lab-based recordings. Leveraging this platform, we collected 50 hours of free-living ExG recordings from 22 participants engaged in unconstrained daily activities. Furthermore, we gathered 20 hours of targeted task-specific data spanning the five human senses (*i.e.*, sight, hearing, taste, touch, and smell), establishing the first benchmark for evaluating model performance across diverse tasks.

Moreover, we propose *Physiology-informed Multi-band Tokenization (PiMT)*, an approach designed to learn task-agnostic ExG representations. Instead of relying on a task-specific narrow band or a single wide-band input, PiMT decomposes ExG data into 12 fixed sub-band tokens, each corresponding to distinct physiological modalities. For instance, the [0.5–4 Hz] band captures EEG delta waves, which are informative for sleep staging (Elsaid & Labanowski, 2017), whereas the [15–45 Hz] band reflects low-frequency EMG activities, relevant for muscle activation and motor tasks (Allison & Fujiwara, 2002). These structured tokens provide the encoder with fine-grained access to diverse spectral features, enabling the model to capture task-relevant information while remaining agnostic to any specific task. Coupled with self-supervised reconstruction objectives, we train a robust, transferable representations that generalize effectively across diverse downstream tasks.

To evaluate our approach, we benchmark PiMT against the state-of-the-art ExG training approaches. Specifically, we evaluate it on our newly introduced DailySense benchmark, which spans tasks across the five human senses, along with four widely used datasets covering diverse ExG applications, including emotion recognition, sleep staging, and brain–computer interface (BCI) tasks. Extensive experiments demonstrate that PiMT achieves robust performance and strong generalization across both DailySense and public datasets. Our key contributions are as follows:

- We identify the key limitations of existing ExG frameworks—insufficient dataset diversity and task-specific model design—that hinder generalization to real-world applications.

- We introduce NeuroBuds, an earphone-based prototype for unobtrusive, long-term ExG monitoring. Leveraging NeuroBuds, we curate DailySense, a dataset containing 50 hours of free-living recordings and 20 hours of task-specific data spanning the five human senses.

- We propose PiMT, a task-agnostic ExG training approach that incorporates a novel, physiology-informed multi-band tokenization scheme. This enables automatic extraction of task-relevant features across the entire frequency spectrum.

- Extensive experiments on DailySense spanning six distinct tasks and four public ExG benchmarks show PiMT achieves state-of-the-art performance, with an average F1-score of 87.6% over baseline models.

Together, these contributions form an integrated co-design in which the hardware, dataset, and training approach are mutually enabling, establishing a scalable foundation for real-world ExG analysis that bridges wearable sensing and foundation models for deeper human understanding.

## 2 RELATED WORK

To enable effective analysis of ExG signals and uncover valuable physiological patterns, recent approaches can be categorized into following three main groups: (i) *Conventional deep learning frameworks*, such as EEGNet (Lawhern et al., 2018) and DeepConvNet (Schirrmeister et al., 2017), leverage temporal and spatial convolutions to extract features directly from raw ExG signals. (ii) *Transformer-based models*, which capture local and long-term temporal dependencies, are well-suited for complex,

**Physiology-informed Multi-band Tokenization (PiMT)**

Figure 1: Overview of PiMT. ExG signals are decomposed into 12 sub-bands via Physiology-informed Multi-band Tokenization (PiMT). A Bidirectional-Mamba encoder processes the tokens, and the model is pre-trained with reconstruction tasks before fine-tuning on downstream tasks.

high-dimensional ExG signals. Early efforts such as EEGConformer (Song et al., 2022) combine convolution and attention to jointly model local and global patterns. PatchTST (Nie et al., 2023) introduces patch-wise attention and independent channel encoding, while Medformer (Wang et al., 2024) enhances feature extraction through multi-scale patching and cross-channel attention. Most recently, Bidirectional-Mamba (Zhu et al., 2024) applies bidirectional state-space modeling for efficient long-range dynamics. (iii) *Self-supervised learning methods* aim to learn generalizable representations from unlabeled ExG signals using proxy tasks such as masked modeling or contrastive learning. BrainBERT (Wang et al., 2023) first applied BERT-style masked modeling to intracranial EEG spectrograms. BIOT (Yang et al., 2023) extends this idea to cross dataset via patch-token transformers, and BrainWave (Yuan et al., 2024) further scaled it to foundation models trained on large clinical datasets. However, despite these advances, existing approaches typically focus on specific ExG tasks and modalities, and are primarily evaluated on lab-controlled datasets. This limitation presents an opportunity to develop more generalized and robust representations from free-living ExG data. We address these gaps by introducing a unified, frequency-agnostic framework trained on real-world data, improving both robustness and practical usability.

## 3 LEARNING TASK-AGNOSTIC ExG REPRESENTATION

**Motivation.** Real-world ExG tasks are often associated with distinct physiological frequency bands. For example, gaze tracking with EOG signals typically relies on low-frequency components in 0.1–15 Hz range (Merino et al., 2010), whereas EEG-based emotion recognition depends on higher-frequency bands, such as 8–30 Hz (Gkintoni et al., 2025). Prior methods either design task-specific models (Gao et al., 2024; Altaheri et al., 2023) or apply narrow-band filters (Farhana et al., 2023; Apicella et al., 2021), both of which limit generalization across tasks. While a wide-band filter (*e.g.*, 0–100 Hz) offers broader coverage, it suffers from loss of physiological features and poor task adaptation. Moreover, different hardware configurations may emphasize or attenuate specific frequency components, motivating a flexible representation that does not rely on prior assumptions about which bands or electrodes are informative. We aim to develop a task-agnostic method that generalizes across tasks without relying on task-specific customization.

**Overview.** We propose a training framework that enables NeuroBuds to generalize effectively across diverse tasks. Figure 1 provides the overview. First, *Physiology-informed Multi-band Tokenization (PiMT)* decomposes the input into 12 physiology-informed sub-bands, producing tokens that grant the model fine-grained access to task-relevant features across different frequency ranges. Next, a

Bidirectional-Mamba encoder generates embeddings from the tokenized representations. To leverage unlabeled free-living data, we introduce a *reconstruction-based pre-training* to learn robust ExG representations. The pre-trained encoder is then fine-tuned on downstream tasks.

## 3.1 Physiology-informed Multi-band Tokenization (PiMT)

To enable the task-agnostic framework, we design a two-step tokenization pipeline that converts raw ExG signals into structured embeddings: (i) physiology-informed frequency decomposition via an *ExG Filter Bank*, and (ii) *Patch Segmentation and Tokenization* to generate input tokens.

**ExG Filter Bank.** Instead of relying on task-specific frequency bands, we design a fixed filter bank grounded in established physiological knowledge of ExG signals (Niedermeyer & Lopes da Silva, 2005; Nunez & Srinivasan, 2006; Task Force, 1996). Concretely, we define 12 canonical sub-band filters spanning key physiological modalities: EEG-delta (0.5–4 Hz), EEG-theta (4–8 Hz), EEG-alpha (8–13 Hz), EEG-beta (13–30 Hz), EEG-gamma (30–100 Hz), EMG-Low-Frequency (15–45 Hz), EMG-Mid-Frequency (45–95 Hz), EMG-High-Frequency (95–100 Hz), EOG-overall (0.1–20 Hz), ECG-Low-Frequency (0.03–0.12 Hz), ECG-High-Frequency (0.12–0.488 Hz), and the QRS complex (8–50 Hz).

**Multi-Band Filtering.** ExG signals are decomposed into complementary spectral components by simultaneously applying all filters in the bank. This decomposition provides the model with fine-grained, physiologically relevant features that span multiple modalities and tasks, rather than forcing reliance on a single-band representation. Formally, given a multi-channel ExG signal $X_c \in \mathbb{R}^T$, where $X_c$ is from channel $c$ over $T$ time steps, we apply the $N_F$ filters to obtain band-specific signals $X_{f,c} \in \mathbb{R}^T$, where $f \in \{1, \ldots, N_F\}$. Each $X_{f,c}$ retains only the components within band $f$, serving as the foundation for subsequent tokenization.

**Patch Segmentation and Tokenization.** The band-specific signal $X_{f,c}$ is segmented into non-overlapping patches, *i.e.*, $\mathbf{p}_{f,c,l} \in \mathbb{R}^w$, where $w$ denotes the patch size and $l$ indexes its temporal position. This segmentation improves computational efficiency and facilitates modeling long-range temporal dependencies (Nie et al., 2023). Together, each patch is contextualized by three dimensions: frequency $f$, channel $c$, and time $l$, forming a structured 3D representation of the ExG input. Finally, each patch $\mathbf{p}_{f,c,l}$ is projected into a latent embedding $e_{f,c,l} \in \mathbb{R}^d$ through a learnable tokenizer, where $d$ denotes the embedding dimension. Specifically, we use a single linear layer shared across all tokens to map each patch into a fixed-dimensional embedding space.

## 3.2 Encoder

We adopt Bidirectional-Mamba for its strong ability to capture long-range sequential dependencies (Schiff et al., 2024; Shams et al., 2024). A detailed analysis of its effectiveness compared with standard Transformers on ExG data is provided in Appendix H. Furthermore, since PiMT introduces an additional frequency dimension that increases sequence length, Mamba is especially suitable: it achieves linear-time complexity in sequential modeling, whereas Transformers suffer from quadratic complexity (Gu & Dao, 2024).

To fully leverage the rich structure of multi-channel ExG signals, we organize the input tokens along three axes, *i.e.*, frequency, channel, and time, in a fixed scanning sequence. Based on empirical validation, we adopt a frequency-first ($f$), channel-second ($c$), and time-last ($l$) ordering scheme. To achieve this, the embeddings $e_{f,c,l}$ are flattened into a sequence following the ($f \times c \times l$) order and then passed into the encoder. The encoder produces contextualized representations $\mathbf{z}$, which are subsequently fed into the downstream task heads.

## 3.3 Pre-Training from Free-Living ExG Data

Most existing ExG models are trained on lab-controlled datasets using task-specific designs, limiting their ability to generalize to real-world conditions. In contrast, ExG signals collected in free-living environments provide richer diversity and broader coverage of human activities, enabling models to learn more robust and general-purpose representations. To exploit this free-living unlabeled data, we adopt a self-supervised pre-training based on reconstruction objectives. Our design choice is motivated by prior work showing that reconstruction outperforms alternatives, *i.e.*, contrastive

learning, when training physiological foundation models on unlabeled data (Narayanswamy et al., 2025). To ensure robust feature extraction, we define six distinct reconstruction tasks, each paired with a dedicated decoder, which are jointly used to train the encoder $E_\theta$.

**Autoencoding:** Given a sequence of patches $\mathbf{p}$ generated from a raw signal $\mathbf{x}$, the encoder $E_\theta$ maps it into a latent representation $\mathbf{z} = E_\theta(\mathbf{p})$. The decoder $D_\phi^{\text{AE}}$ reconstructs the original signal $\hat{\mathbf{p}}^{\text{AE}} = D_\phi^{\text{AE}}(\mathbf{z})$. This task encourages the encoder to capture temporal features while reducing noise.

**Masked Reconstruction:** To enforce contextual learning, we employ masked reconstruction (Devlin, 2018; He et al., 2022). The patches $\mathbf{p}$ are partially masked along time, channel, and frequency dimensions, producing a corrupted version $\mathbf{p}^{\text{mask}}$. The encoder processes the masked input to yield $\mathbf{z}^{\text{mask}} = E_\theta(\mathbf{p}^{\text{mask}})$. The decoder $D_\phi^{\text{MR}}$ recovers the original signal, generating $\hat{\mathbf{p}}^{\text{MR}} = D_\phi^{\text{MR}}(\mathbf{z}^{\text{mask}})$.

**Frequency Domain Feature Reconstructions:** To capture spectral information, we incorporate two frequency-domain reconstruction tasks. We first apply the Fast Fourier Transform (FFT) to obtain amplitude $\mathbf{p}^{\text{A}}$ and phase $\mathbf{p}^{\text{P}}$. Two decoders, $D_\phi^{\text{A}}$ and $D_\phi^{\text{P}}$, reconstruct these features from the encoded representation $\mathbf{z}$, producing $\hat{\mathbf{p}}^{\text{A}}$ and $\hat{\mathbf{p}}^{\text{P}}$, respectively. Specifically, the two decoders aim to recover the original frequency domain signals based on: $\hat{\mathbf{p}}^{\text{A}} = D_\phi^{\text{A}}(\mathbf{z})$ and $\hat{\mathbf{p}}^{\text{P}} = D_\phi^{\text{P}}(\mathbf{z})$, respectively.

**Masked Frequency Domain Reconstructions:** To enhance the model's capacity to infer spectral features from incomplete inputs, we apply the same frequency reconstruction tasks to masked input signals. Two additional decoders, $D_\phi^{\text{MA}}$ and $D_\phi^{\text{MP}}$, reconstruct the amplitude and phase, producing $\hat{\mathbf{p}}^{\text{MA}}$ and $\hat{\mathbf{p}}^{\text{MP}}$ from $\mathbf{z}^{\text{mask}}$. The new decoders aim to recover the original frequency domain signals based on: $\hat{\mathbf{p}}^{\text{A}} = D_\phi^{\text{MA}}(\mathbf{z}^{\text{mask}})$ and $\hat{\mathbf{p}}^{\text{P}} = D_\phi^{\text{MP}}(\mathbf{z}^{\text{mask}})$, respectively.

To jointly optimize the self-supervised objectives, we assign each task an independent decoder that reconstructs a specific aspect of the input signal, while sharing the encoder. Training is guided by mean absolute error (MAE) losses between the original and reconstructed signals. We combine these losses into a single objective by weighting each task-specific loss with a coefficient $\lambda$, which controls its relative contribution. The overall reconstruction loss is thus a weighted sum across all tasks. Each decoder is implemented as a lightweight MLP designed to reconstruct the target sequence. The $\lambda$ values were empirically selected, and details are provided in Appendix G.

### 3.4 Fine-Tuning

Building on the representations learned from free-living data, we fine-tune the model to diverse downstream tasks (*e.g.*, sight, hearing, taste, touch, and smell). The pre-trained encoder serves as a feature extractor, while task-specific decoders are trained on labeled data. For classification tasks, we aggregate the encoder's patch-wise outputs into a fixed-length feature vector via mean pooling. The vector is then passed through a fully connected classification decoder trained with cross-entropy loss. For continuous regression tasks, such as gaze tracking, we employ a linear decoder operating at the patch level to generate sequential outputs, which are then aggregated into the final prediction. The model is optimized using a standard regression loss.

## 4 DailySense: Free-Living ExG Data across Five Human Senses

We built *DailySense*, an ExG dataset collected through earphones, designed to enhance the ExG dataset diversity beyond traditional lab-controlled settings and to enable benchmarking across a broad spectrum of daily life tasks. DailySense includes data from 22 participants, including 50 hours of unlabeled free-living recordings and 20 hours of labeled task-specific data spanning the five fundamental human senses.[1] Compared with existing lab-based ExG benchmarks, which often involve a similar number of participants but shorter recording durations (*e.g.*, DREAMER includes 23 participants with approximately 20 hours of data), DailySense provides a more diverse and comprehensive dataset, laying a stronger foundation for generalizable ExG representation learning.

**Data Collection Platform.** To collect free-living ExG data, we developed *NeuroBuds*, an earphone-integrated ExG sensing prototype. Unlike traditional head-mounted ExG devices that are bulky and

---

[1]Our data collection was approved by the Institutional Review Board (IRB). We are also currently working with our legal team to determine the possibility of publicly or conditionally sharing the dataset.

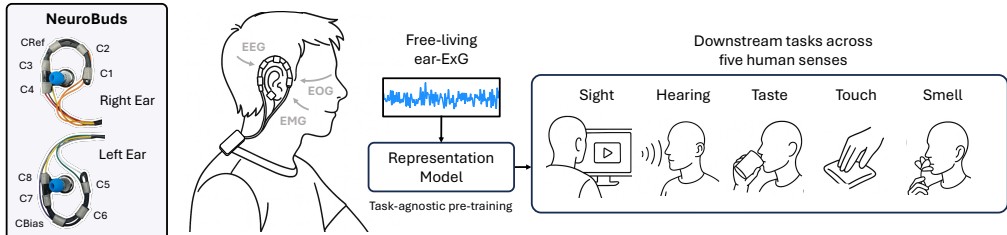

Figure 2: Overview of *DailySense* dataset. Using our earphone-based ExG analysis device, NeuroBuds, we collect free-living unlabeled data for task-agnostic pre-training and labeled data spanning five human senses, serving as benchmarks for downstream tasks.

expensive ($10,000–$50,000), NeuroBuds employs an earhook-style form factor that is low-cost and compact, and well-suitable for scalable, long-term daily use. The device integrates amplification, digitization, onboard storage, and wireless transmission into a lightweight PCB ($4.2\,\mathrm{cm} \times 2.2\,\mathrm{cm}$, 20 g, $80). During data collection, participants wore earphones with integrated electrodes and carried the PCB as shown in Figure 2. The around-the-ear electrodes can then capture ExG signals, including EEG (sites T7–T10, FT7–FT10, TP7–TP10), auricular electrodes for EMG, and lateral electrodes for EOG, providing cognitive, muscular, and ocular coverage. We detail hardware design in Appendix A, and the physiological rationale behind the design and signal quality for each modality in Appendix B.

**Data Collection Protocol.** DailySense contains (i) unlabeled data of daily life and (ii) labeled data spanning the five human senses. A total of 22 participants (ages 23–62, 16 men, 6 women) wore NeuroBuds during daily routines without restrictions, performing natural activities such as walking, eating, talking, and facial movements. This produced 50 hours of free-living ear-ExG recordings. Furthermore, we curated six benchmark tasks covering the five human senses: (i) gaze tracking, (ii) interest inference while watching videos (sight), (iii) interest inference while listening to audio (hearing), (iv) surface texture classification (touch: rough vs. smooth), (v) taste classification (sweet vs. sour), and (vi) smell classification (floral vs. sour). Data were collected in a task-controlled environment with up to seven participants per task, producing 20 hours of labeled recordings. All experimental protocols followed prior brain-computer interface studies (Amini et al., 2022; Iravani et al., 2019; Namazi & Kulish, 2016; Vo et al., 2023; Xia et al., 2023). Further experimental details are provided in Appendix C.

**Data Processing.** We adopted a minimal pre-processing pipeline to ensure fair and uniform comparison across methods. While certain tasks may benefit from additional task-specific filtering or explicit frequency-domain transforms, introducing such pre-processing would encode task-dependent assumptions that conflict with our goal of task-agnostic modeling. Following established protocols (Jiang et al., 2024), each channel's time-domain signal first undergoes second-order IIR notch filtering (quality factor $Q = 30$) at both 50 Hz and 60 Hz using zero-phase forward–backward filtering. The notch-filtered signal is then processed using band-pass filtering implemented with first-order Butterworth low-pass and high-pass filters, again applied with zero-phase filtering. When PiMT is applied, 12 filter banks are constructed as parallel branches from independent copies of the notch-filtered signal. Each branch corresponds to a predefined physiological frequency band and serves as an isolated physiological view of the same underlying signal. After filtering, signals are resampled to 200 Hz and normalized. The continuous recordings are segmented into non-overlapping 4-second windows. To improve model robustness, we augment the training data with small additive noise. Appendix D provides visualizations of the collected signals.

## 5 EXPERIMENTS

### 5.1 EXPERIMENTAL SETUP

**Baselines.** We benchmarked PiMT against baselines including a traditional machine learning model (SVM), ExG-specific neural architectures (DeepConvNet (Schirrmeister et al., 2017), EEGNet (Lawhern et al., 2018), and EEGConformer (Song et al., 2022)), and general-purpose time-series models (Time-Series Transformer (TST) (Zerveas et al., 2021), PatchTST (Nie et al., 2023), and Bidirectional-Mamba (Zhu et al., 2024)). Among them, we emphasize PatchTST as a strong baseline—an advanced

Table 1: Performance of PiMT and baselines on DailySense. Classification results are in F1-score, and gaze tracking performance is in angular error. Best results are highlighted in **bold**.

| Method | Classification (↑) | | | | | | Regression (↓) |
| | Video | Audio | Taste | Touch | Smell | Avg. | Gaze |
|---|---|---|---|---|---|---|---|
| *Without pre-training* | | | | | | | |
| SVM | $0.665 \pm 0.078$ | $0.610 \pm 0.126$ | $0.556 \pm 0.114$ | $0.554 \pm 0.107$ | $0.510 \pm 0.084$ | 0.579 | $6.60° \pm 1.27°$ |
| EEGNet | $0.753 \pm 0.137$ | $0.712 \pm 0.149$ | $0.709 \pm 0.088$ | $0.643 \pm 0.097$ | $0.669 \pm 0.063$ | 0.697 | $6.52° \pm 1.24°$ |
| DeepConvNet | $0.680 \pm 0.174$ | $0.706 \pm 0.129$ | $0.633 \pm 0.074$ | $0.638 \pm 0.075$ | $0.636 \pm 0.062$ | 0.659 | $7.04° \pm 1.31°$ |
| TST | $0.773 \pm 0.125$ | $0.705 \pm 0.104$ | $0.731 \pm 0.068$ | $0.669 \pm 0.116$ | $0.667 \pm 0.096$ | 0.709 | $6.54° \pm 1.30°$ |
| PatchTST | $0.771 \pm 0.146$ | $0.749 \pm 0.113$ | $0.731 \pm 0.092$ | $0.686 \pm 0.119$ | $0.681 \pm 0.049$ | 0.724 | $6.47° \pm 1.28°$ |
| EEGConformer | $0.738 \pm 0.127$ | $0.752 \pm 0.141$ | $0.688 \pm 0.062$ | $0.678 \pm 0.102$ | $0.670 \pm 0.047$ | 0.705 | $6.53° \pm 1.28°$ |
| Bidirectional-Mamba | $0.820 \pm 0.102$ | $0.858 \pm 0.113$ | $0.733 \pm 0.060$ | $0.762 \pm 0.101$ | $0.722 \pm 0.067$ | 0.779 | $6.53° \pm 1.16°$ |
| **PiMT (Ours)** | $\mathbf{0.858} \pm \mathbf{0.084}$ | $\mathbf{0.885} \pm \mathbf{0.125}$ | $\mathbf{0.790} \pm \mathbf{0.077}$ | $\mathbf{0.807} \pm \mathbf{0.113}$ | $\mathbf{0.753} \pm \mathbf{0.069}$ | **0.819** | $\mathbf{6.11°} \pm \mathbf{1.20°}$ |
| *With pre-training* | | | | | | | |
| PatchTST | $0.807 \pm 0.146$ | $0.786 \pm 0.146$ | $0.697 \pm 0.099$ | $0.700 \pm 0.131$ | $0.670 \pm 0.082$ | 0.732 | $6.42° \pm 1.33°$ |
| **PiMT (Ours)** | $\mathbf{0.964} \pm \mathbf{0.028}$ | $\mathbf{0.961} \pm \mathbf{0.038}$ | $\mathbf{0.801} \pm \mathbf{0.064}$ | $\mathbf{0.860} \pm \mathbf{0.118}$ | $\mathbf{0.793} \pm \mathbf{0.069}$ | **0.876** | $\mathbf{6.00°} \pm \mathbf{1.13°}$ |

masked reconstruction model built on a Transformer backbone that independently models each ExG channel and excels at capturing long-range temporal modeling.

**Benchmark Datasets.** We evaluated PiMT on DailySense along with four widely used ExG benchmarks: DREAMER (Katsigiannis & Ramzan, 2018) and SEED (Zheng & Lu, 2015) for emotion recognition, Sleep-EDF (Kemp et al., 2000) for sleep stage classification, and BCI Competition IV 2b (Leeb et al., 2008) for motor imagery. Dataset details are provided in Appendix E.

**Implementation Details and Metrics.** Our implementation consists of two primary stages: (i) pre-training on free-living data and (ii) task-specific fine-tuning. We first pre-train PiMT on 50 hours of free-living data using a masked reconstruction objective, then fine-tune it on each downstream dataset using an 8:2 train-test split for each participant. For evaluation, we report the mean squared error (MSE) in angular (°) units for gaze tracking and macro-averaged F1-scores for all classification tasks. All tasks are repeated three times with different random seeds, and we report the corresponding standard deviation. For public benchmarks, data preprocessing follows the official EEGConformer implementation. Additional training and reproducing details, resource specifications, and hyperparameter tuning are provided in Appendix F and Appendix G.

## 5.2 EVALUATION ON DAILYSENSE

Table 1 shows the F1-scores of PiMT compared with the baselines on DailySense. Overall, the results demonstrate that ear-ExG combined with PiMT can effectively capture five human senses, achieving up to 81.9% F1-score and as low as 6.11°gaze error, even without pre-training. Notably, PiMT outperformed all baselines, achieving a 4% improvement in F1-score and a 0.41°reduction in gaze error. We attribute this generalizability to PiMT's ability to interpret task-relevant frequency bands, a capability essential for handling diverse ExG-based tasks characterized by heterogeneous frequency-band features. We also observed that the Mamba-based backbone contributed significantly to performance gains; detailed comparisons against Transformer-based variants are in Appendix H.

**Effect of Pre-Training on Free-Living Data.** A key advantage of NeuroBuds is its ability to facilitate effortless collection of ExG signals, enabling large-scale pre-training. We evaluated the performance of PiMT when pre-trained on free-living data and compared it with PatchTST, which is the only baseline with a tailored pre-training strategy. As shown in Table 1, pre-training improved the average F1-score of PiMT from 81.9% to 87.6%. Similarly, PatchTST improved from 72.4% to 73.2%, whereas PiMT demonstrated a substantially larger gain. These results highlight the effectiveness of both our hardware-enabled free-living data collection and our reconstruction-based pre-training framework. Further analysis of our pre-training is provided in Appendix I and Appendix J.

## 5.3 EVALUATION ON PUBLIC BENCHMARKS

To validate the generalizability of PiMT beyond the *DailySense* dataset, we evaluated it on four widely used public benchmarks covering diverse ExG tasks. We compared against two strongest

Table 2: F1-score on four public ExG benchmarks across various tasks.

| Baselines | DREAMER | SEED | Sleep-EDF | BCI Competition IV 2b |
|---|---|---|---|---|
| PatchTST | $0.889 \pm 0.085$ | $0.756 \pm 0.093$ | $0.810 \pm 0.005$ | $0.657 \pm 0.008$ |
| Bidirectional-Mamba | $0.875 \pm 0.090$ | $0.750 \pm 0.107$ | $0.796 \pm 0.002$ | $0.646 \pm 0.015$ |
| **PiMT (Ours)** | $\mathbf{0.910 \pm 0.074}$ | $\mathbf{0.820 \pm 0.121}$ | $\mathbf{0.822 \pm 0.006}$ | $\mathbf{0.693 \pm 0.004}$ |

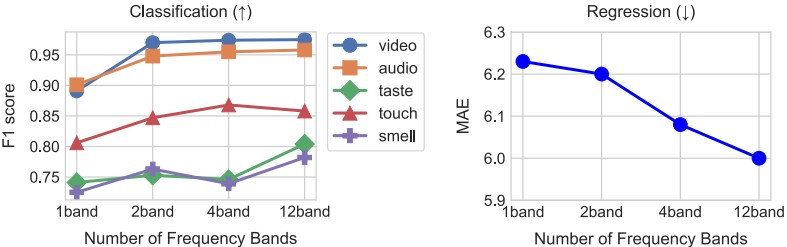

Figure 3: Comparison of different ExG tokenization strategies: 1-band (0.1–75 Hz), 2-band (0.1–15 Hz and 15–75 Hz), 4-band (0.1–5 Hz, 5–15 Hz, 15–35 Hz, and 35–75 Hz), and our 12-band filter bank (described in Section 3.1).

baselines, PatchTST (Nie et al., 2023) and Bidirectional-Mamba (Zhu et al., 2024). As shown in Table 2, PiMT consistently outperformed all baselines across all datasets. Overall, these results demonstrate that our training strategy learns general-purpose ExG representations through PiMT, leading to robust performance across diverse benchmarks. This confirms generalization beyond the self-collected DailySense dataset to real-world benchmarks, which underscores the potential of NeuroBuds as a unified framework for generalizable ExG representation learning.

## 5.4 EFFECT OF MULTI-BAND TOKENIZATION

To understand the impact of multi-band tokenization, we compared the model performance under different frequency-band tokenization strategies on DailySense. As shown in Figure 3, performance consistently improves as the number of bands increases. Our 12-band filter bank approach outperforms the 1-, 2-, and 4-band variants, achieving an average 4.6% F1-score gain on classification tasks and the lowest gaze-tracking error. These results suggest that fine-grained decomposition allows NeuroBuds to exploit subtle but physiologically meaningful spectral cues. We also compare PiMT with common time–frequency tokenization methods in Appendix K, demonstrating its superior performance.

**Saliency Analysis.** To further understand the impact on the downstream task, we conducted a visual analysis of how different frequency bands contribute to each task. Figure 4 depicts saliency maps that highlight the contribution of each frequency-band token during inference. Importantly, we observed clear task-relevant activation patterns: (i) gaze and video tasks, which are closely linked to eye movements, exhibited strong activation in low-frequency bands (Plöchl et al., 2012), and (ii) touch, taste, smell, and auditory interest classification emphasized high-frequency components, consistent with somatosensory beta–low-gamma activity involved in processing external stimuli (Bauer et al., 2006) and peri-auricular EMG spectra reflecting near-ear muscle movements (Goncharova et al., 2003). These findings demonstrate that PiMT enables the model to dynamically focus on task-relevant frequency components without explicit supervision. We stress that this property is essential for enabling generalizable ExG-based applications in daily-life scenarios using NeuroBuds.

## 5.5 IMPACT OF PRE-TRAINING DATA SCALE

We examine how the scale of pre-training data influences representation quality and downstream task performance. To this end, we randomly split the pre-training corpus into 80% training and 20% held-out test data, and subsampled varying proportions of the training set. Figure 5 shows the test loss across epochs under varying training data scales. As expected, larger pre-training sets consistently produced lower losses, indicating that PiMT benefits from additional data and scales effectively.

Figure 6 presents downstream results on DailySense. For classification tasks, average performance saturates around 30% of the pre-training data, suggesting diminishing returns beyond this point.

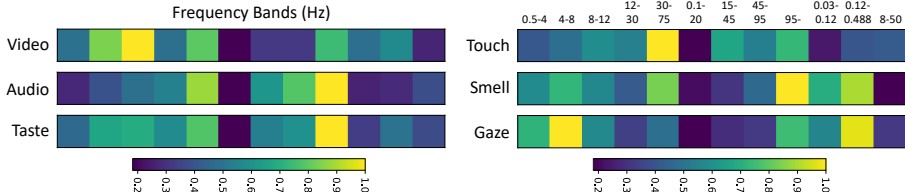

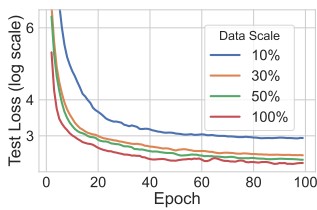

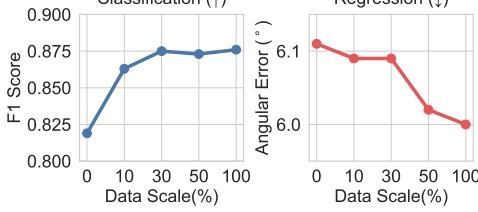

Figure 4: Saliency analysis demonstrating how the model dynamically captures task-relevant frequency bands via multi-band tokenization.

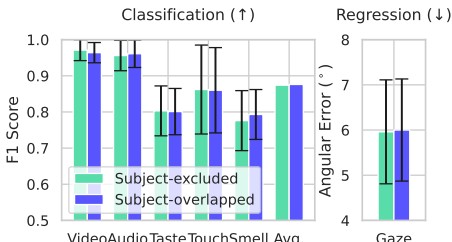

Figure 5: Test loss across different pre-training data scales.

Figure 6: Downstream performance of PiMT with varying amounts of pre-training data.

In contrast, gaze regression continues to improve up to 50%, highlighting task-dependent benefits of larger pre-training scales. Overall, these findings suggest that while some tasks quickly reach saturation, others continue to benefit from larger-scale pre-training. Importantly, the consistent loss reductions in Figure 5 confirm that PiMT can effectively exploit additional data, underscoring its promise as a general-purpose ExG representation model.

## 5.6 FURTHER ANALYSIS AND ABLATION STUDY

**On-Device Deployment and Real-Time Analysis (Appendix N)**. We evaluated the runtime overhead of PiMT (with Transformer as a backbone) on a commercial smartphone (Samsung Galaxy S24), which serves as a representative companion device for earphones. The model achieved efficient runtime performance with an average inference latency of 25 ms, memory usage of 266 MB (3.6%), and CPU utilization of 20.3%.

**Leave-One-Subject-Out Evaluation**. To further assess cross-participant generalization, we conducted leave-one-subject-out (LOSO) experiments on DailySense. In DailySense, some participants contributed to both free-living pre-training data and task-specific data. Therefore, in LOSO, for each target user, their data was excluded from pre-training and used only for testing. Despite this constraint, PiMT achieved performance comparable to full pre-training (Figure 7), confirming its ability to generalize effectively to unseen users.

Figure 7: LOSO Performance when the target subject's data is either included or excluded.

**Ablation Study (Appendix J, L)**. We performed an ablation study to investigate the contribution of each pre-training component to representation learning. As shown in Table 7, the complete model achieved the highest overall performance. We observed consistent performance improvements as additional components were incorporated, indicating the complementary benefits of each module. Furthermore, Table 9 shows that a patch size of 0.5 seconds yielded the best downstream performance compared to alternative configurations.

**Comparison with Controlled Pre-Training Datasets (Appendix Q).** We compare pre-training on our 50-hour DailySense dataset against larger controlled corpora: TUAR (Hamid et al., 2020) (98.6 hours) and TUAR+TUSZ (Shah et al., 2018) (498.6 hours). Despite its smaller size, DailySense achieves the best classification performance with competitive gaze accuracy, demonstrating that free-living data yields more transferable representations than controlled recordings.

# 6 DISCUSSION

We acknowledge several limitations of our current approach. Like many existing ExG frameworks, the scale of the collected dataset remains relatively limited. Although DailySense includes 50 hours of free-living recordings and 20 hours of task-specific data from 22 participants, comparable to or exceeding many existing lab-based ExG benchmarks, it is still modest by the standards of large-scale wearable sensing studies. In addition, the demographic diversity of the participants is limited, which may restrict the model's ability to generalize across broader populations. Our LOSO experiments show promising cross-subject generalization; however, performance drops when training and testing on different users (Appendix O) highlight the persistent challenge of inter-subject variability (*e.g.*, individual physiology and sensor placement). Addressing full cross-subject generalization is beyond the primary scope of this work, which focuses on task-agnostic representation learning under mixed ExG signals. Nonetheless, by demonstrating the effectiveness of earphone-derived free-living ExG data for representation learning, NeuroBuds provides a scalable path toward broader population-level data collection and establishes the foundation for a more generalizable framework in future work.

# 7 CONCLUSION

We tackle two long-standing barriers in ExG analysis: (i) the lack of diverse, real-world data and (ii) the reliance on task-specific model designs. To address data diversity, we developed NeuroBuds, an earphone-based sensing prototype, and curated DailySense, the first dataset with 50 hours of free-living and 20 hours of task-specific ExG data spanning all five human senses. To overcome task-specificity, we propose Physiology-informed Multi-band Tokenization (PiMT), which decomposes ExG signals into structured tokens across 12 canonical sub-bands aligned with distinct physiological modalities. Combined with reconstruction-based pre-training on free-living data, PiMT learns robust representations that generalize across tasks. Evaluations on DailySense and four public benchmarks demonstrate that PiMT consistently outperforms state-of-the-art baselines. Together, these contributions push ExG research beyond narrow, lab-constrained applications toward generalizable and real-world physiological sensing. Looking ahead, this work opens new opportunities in personalized health monitoring, cognitive interfaces, and scalable everyday sensing powered by ExG platforms.

## ACKNOWLEDGMENTS

This work was supported by the Institute of Information & communications Technology Planning & Evaluation (IITP) grant funded by the Korea government (MSIT) (No.2024-00444862, Non-invasive near-infrared based AI technology for the diagnosis and treatment of brain diseases). We sincerely thank Zilong Wang, Jiazhao Wang, Legend Zhu, Nicholas Yuan, and Qi Zhang for their insightful discussions and strong support throughout this project.

## ETHICS STATEMENT

Our data collection was approved by the Institutional Review Board (IRB), ensuring the safety of both the participants and the device prototype used in the study. For the other experiments, we used publicly available datasets, which were used in accordance with their intended purposes. There is no ethical issue with this paper.

## REPRODUCIBILITY STATEMENT

Our Physiology-informed Multi-band Tokenization approach can be reproduced using the filter bank described in Section 3.1. Experimental details are provided in Section 5, Appendix F and G.

## USAGE OF LARGE LANGUAGE MODELS

Large Language Models (LLMs) were used solely for polishing the writing of this paper.

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

## A  NEUROBUDS HARDWARE DESIGN

To enable large-scale, in-the-wild ExG data collection, we built an earphone-based sensing platform consisting of two main components:

**Earphone-Shaped Sensing Array:** To adopt an earhook-style form factor, we use a commercial earphone (Powerbeats PB123) as the backbone, and wrap conductive tape around the frame to form electrodes. Each side includes five electrodes: the top ones on the left and right act as bias and reference, while the remaining eight serve as signal channels.

**Lightweight Processing Board:** We design a custom Printed Circuit Board (PCB) integrating signal amplification, digitization, wireless transmission, and onboard storage:

- **Amplification:** An bio-amplifier chip (ADS1299) and the front-end circuit support 8-channel ExG signal conditioning.

- **Digitization and Control:** An ESP32 microcontroller handles A/D conversion, peripheral control, and real-time streaming.

- **Wireless Streaming:** Microcontroller's built-in Wi-Fi/BLE enables direct transmission to phones or PCs for data collection or real-time on-device inference.

- **Storage:** A microSD slot supports continuous onboard logging.

To minimize size and weight without compromising signal integrity, we adopted highly integrated chips (ADS1299, ESP32), and designed a compact 6-layer PCB with dense layout of components on both sides to further reduce footprint. The resulting design measures just $4.2\,\text{cm} \times 2.2\,\text{cm}$ and weighs only $20\,\text{g}$, which is significantly smaller than existing COTS systems like OpenBCI ($6.1\,\text{cm} \times 6.1\,\text{cm}$, $80\,\text{g}$) or OpenEarable ($5.7\,\text{cm} \times 3\,\text{cm}$, only support 2 ExG channel).

During use, the board is enclosed in a 3D-printed case and connected to the sensing array via Dupont wires. Users can wear the platform unobtrusively during daily activities, with the board placed in a pocket or worn on the body, enabling free-living data collection.

## B  QUALITY OF EEG, EOG, AND EMG SIGNAL

Our electrode placement around the ear was carefully designed to capture EEG, EOG, and EMG signals while maintaining a compact and unobtrusive form factor. Below, we outline the physiological rationale and supporting evidence for each modality.

**EEG:** The electrodes align with standard around-the-ear EEG channels, *i.e.*, T7–T10, FT7–FT10, and TP7–TP10 in the 10-10 EEG system (Seeck et al., 2017). The strong classification performance on cognitive tasks demonstrates that our recordings contain reliable EEG activity.

**EMG:** Electrodes positioned on auricular muscles capture EMG signals linked to facial expressions. We further validated this by recording deliberate facial movements (*e.g.*, blinking, biting), which produced distinct EMG-specific patterns.

**EOG:** Electrodes placed on both sides of the head enable strong horizontal EOG capture, with partial vertical EOG sensitivity due to vertical displacement. Eye movement patterns ($0.1$–$5\,\text{Hz}$) are clearly observed in Appendix D, and our gaze tracking accuracy (within 6.15 degrees as shown in Appendix J) further supports the presence of robust EOG signals.

This electrode configuration enables simultaneous acquisition of EEG, EMG, and EOG signals, providing a rich multimodal ExG dataset while preserving wearability for daily use.

## C  DAILYSENSE DATA COLLECTION PROTOCOL

As the first unified earphone-based system to explore whether minimally intrusive ear-ExG signals can reflect information for human senses, our dataset design strictly follows established brain-sensing protocols traditionally collected by EEG headsets which are often binary settings. In this section, we describe the detailed protocol of our six task-specific experiments. The experimental tasks included:

- **Gaze Tracking:** Participants were seated 60 cm from a 13.5-inch laptop (model: Surface Book 2, display size: $3000 \times 2000$ pixels, vertical refresh rate: 59 Hz). This task evaluated whether EXG signals could accurately track gaze positions. The error was quantified as the angular difference between the ExG-based gaze estimation and the ground truth obtained from a Tobii eye tracker (tob, 2016) (model: Tobii 4C Eye Tracker).

- **Auditory and Video Interest Inference:** Inspired by SEED and DREAMER datasets (Zheng & Lu, 2015; Katsigiannis & Ramzan, 2018), this experiment explored the correlation between ExG signals and engagement with visual/auditory stimuli. Participants were asked to watch or listen to video clips. After each session, they rated their interest level. Each participant watched/listened to six stimulus clips, each lasting six minutes. The goal is to classify the participant's emotional state every four seconds based on ExG responses.

- **Surface Texture Classification (Touch Perception):** Participants interacted with different textured surfaces to analyze ExG responses to tactile stimuli (Amini et al., 2022). Each participant rubbed either a rough or smooth surface for 60 continuous seconds, repeating this process 10 times for each texture. The goal is to classify the participant's touch perception every four seconds.

- **Taste Classification:** This experiment assessed ExG responses to different taste profiles (sweet vs. sour). Participants sipped a liquid and held it in their mouth for 20 seconds (Vo et al., 2023; Xia et al., 2023). To prevent cross-contamination, a 30-second rest period was enforced between different taste samples, allowing participants to rinse their mouths before proceeding to the next task. The task aims to classify the participant's taste perception every four seconds.

- **Smell Classification:** This task examined ExG signal responses to olfactory stimuli (Iravani et al., 2019; Namazi & Kulish, 2016). Participants inhaled pleasant and unpleasant odors, and the model was evaluated on its ability to distinguish between different scent categories.

Table 3 provides a comprehensive summary of the classification labels, stimulus materials, trial durations, number of sessions, and trial structures per participant for each task.

Table 3: Experimental task details.

| Task | Labels | Materials | Trial duration time | Total sessions | Total time | Rest time |
|------|--------|-----------|---------------------|----------------|------------|-----------|
| Taste | Sweet | Chocolate milk | 20 seconds | 15 | 300 seconds | 30 seconds |
|       | Sour | Vinegar | 20 seconds | 15 | 300 seconds | 30 seconds |
| Touch | Rough | Scent paper | 1 minute | 10 | 10 minutes | 20 seconds |
|       | Smooth | Silk | 1 minute | 10 | 10 minutes | 20 seconds |
| Smell | Lavender | Lavender scent bag | 20 seconds | 15 | 300 seconds | 30 seconds |
|       | Sour | Vinegar | 20 seconds | 15 | 300 seconds | 30 seconds |
| Video | Interesting | Comedy Clips | 5 minutes | 6 | 30 minutes | 30 seconds |
|       | Not-interesting | Lectures/Documentary | 5 minutes | 6 | 30 minutes | 30 seconds |
| Audio | Interesting | Comedy podcast | 5 minutes | 6 | 30 minutes | 30 seconds |
|       | Not-interesting | Lectures | 5 minutes | 6 | 30 minutes | 30 seconds |

## D    VISUALIZATION OF EXG SIGNALS

We visualized the raw ExG signals measured using NeuroBuds and illustrated how they are transformed into multi-band tokens through bandpass filtering across different frequency ranges. Figure 8 shows the decomposition of the raw signal into twelve frequency bands, each of which is subsequently tokenized as part of our multi-band sequence. For gaze tracking, low-frequency bands (*e.g.*, EOG-overall and ECG-HF bands) exhibit clearer temporal patterns that align with EOG signals (Merino et al., 2010). In contrast, for tasks such as touch, low-frequency activity is less prominent, while informative features emerge in higher-frequency bands (Manfredi et al., 2014; Kramer et al., 2020).

In addition to evaluating ExG quality implicitly through downstream task performance, we also conducted a direct quantitative comparison between our earphone-based NeuroBuds prototype and a research-grade OpenBCI device. Specifically, we ran an eye-movement tracking experiment with two participants (approximately one hour of synchronized data) and computed Pearson correlations between NeuroBuds and OpenBCI channels. The average cross-system correlation reached 0.71 (statistically significant, $p < 0.001$), demonstrating that NeuroBuds capture ExG/EOG activity with high consistency relative to a laboratory-grade system. Beyond quantitative metrics, we also performed visualization analysis to assess overall signal similarity. Visual comparison of synchronized raw ExG signals shows that NeuroBuds and OpenBCI exhibit closely aligned temporal patterns, with similar waveform shapes, amplitudes, and drift trends throughout the recording as shown in Figure 9. These qualitative observations, together with the correlation analysis, further confirm that NeuroBuds produces ExG signals that closely match those from research-grade devices.

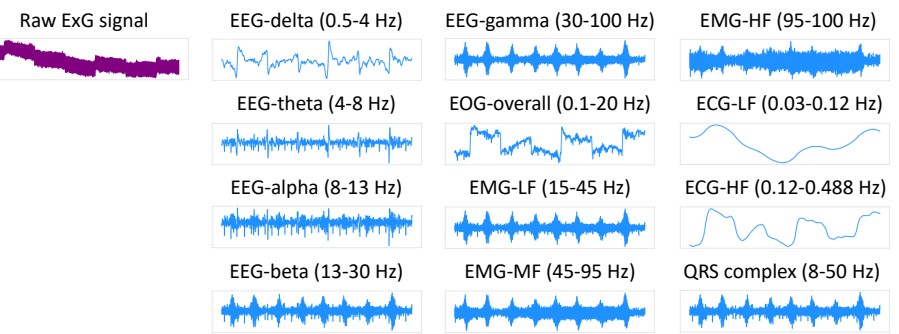

Figure 8: Raw ExG signals from DailySense dataset and their decomposition into twelve physiology-informed frequency bands for PiMT.

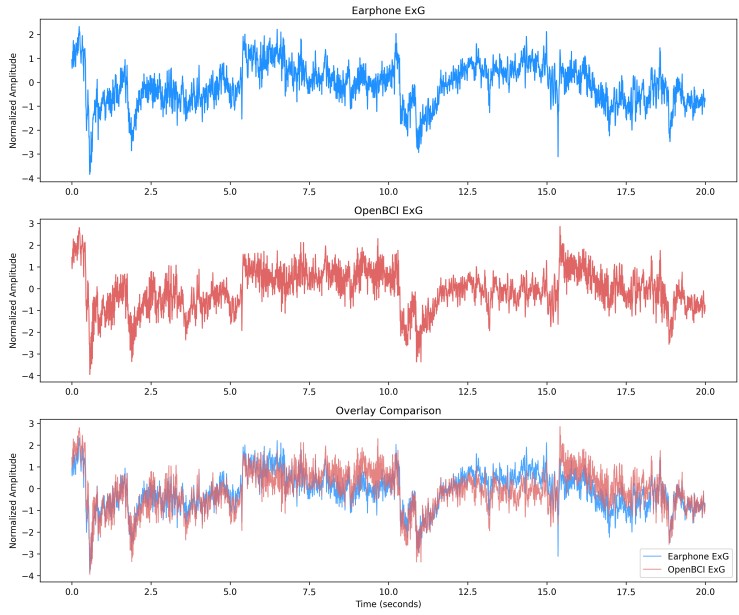

Figure 9: Visualization of synchronized ExG signals collected from NeuroBuds and a research-grade OpenBCI device with the Pearson correlation of 0.7586, demonstrating strong cross-system signal similarity.

Table 4: Statistics of DailySense and benchmark datasets

| Dataset | Device | Setting | Signals | Participants | Hours |
|---|---|---|---|---|---|
| DailySense | Earphone | Daily free-living | ExG | 22 | 70 |
| DREAMER | Headset | Film watching | EEG | 23 | 23 |
| SEED | Headset | Film watching | EEG | 15 | 45 |
| Sleep-EDF | Polysomn. | Sleep lab | EEG,EOG,EMG | 15 | 45 |
| BCI Competition IV 2b | Headset | Controlled | EEG, EOG | 9 | 6 |

## E    BENCHMARK DATASETS

We used four public benchmark datasets to further validate effectiveness of PiMT. The statistics of public benchmark compared with DailySense is demonstrated in Table 4 For all benchmark datasets, we performed a random 80/20 split, assigning 80% of the data to training and the remaining 20% to testing. We followed established protocols (Jiang et al., 2024) to preprocess the ExG signals.

**DREAMER** (Katsigiannis & Ramzan, 2018) is an EEG-based emotion recognition dataset collected from 23 participants while they watched 18 film clips designed to elicit different affective states. The dataset provides signals from electroencephalogram (14 channels at 128 Hz) and electrocardiogram (2 channels at 256 Hz). Each trial is annotated with self-reported valence, arousal, and dominance scores on a 5-point scale. We used the EEG recordings for classification of emotional states, formulating the task as binary classification based on dominance levels, where trials with dominance $\geq 3$ were labeled as high and those with dominance $< 3$ as low.

**SEED** (Zheng & Lu, 2015) is a emotion recognition dataset with EEG recordings from 15 subjects. Participants watched 15 film clips (five positive, five neutral, and five negative) across three sessions. EEG was recorded from 62 channels using the ESI NeuroScan system at 1000 Hz. We used the downsampled signal at 200 Hz. The dataset provides trial-level emotion labels (positive, neutral, negative).

**Sleep-EDF** (Kemp et al., 2000) is a dataset used for sleep stage classification. It contains 197 whole-night polysomnographic recordings from both healthy subjects and patients with mild sleep difficulties. The EEG signals were recorded from two channels (Fpz–Cz and Pz–Oz) at 100 Hz, and the EOG signals were also sampled at 100 Hz. We used five-class scoring (W, N1, N2, N3, REM) for classification only using EEG signals.

**BCI Competition IV 2b** (Leeb et al., 2008) is a motor imagery dataset consisting of EEG recordings from 9 subjects across 5 sessions. Subjects were asked to perform left-hand and right-hand motor imagery tasks. Each session included multiple runs of motor imagery trials, with EEG recorded from three bipolar channels (C3, Cz, C4) at 250 Hz. The dataset defines two classes corresponding to left-hand and right-hand motor imagery.

## F    IMPLEMENTATION DETAILS

For the Bidirectional-Mamba model, we used 8 layers with a hidden state size of 16 and an embedding dimension of 64. The decoder consisted of two fully connected layers, each with a hidden dimension of 64. For the baseline implementations, we tuned SVM using a grid search over $C \in \{0.1, 1, 10, 100\}$, $\gamma \in \{0.01, 0.001, 0.0001\}$, and kernel types (*rbf*, *linear*, *poly*). For the other baselines, we followed their official implementations and performed grid searches to tune key hyperparameters, such as learning rate and batch size.

For the train/test split, we first segmented long sequences into 4-second windows and randomly shuffled them. We then applied a standard 80/20 division to construct the training and test sets.

To address potential inconsistencies with prior work, we replicated the exact data processing pipeline and code from the EEGConformer repository, including resampling, normalization, and standard segmentation protocols. We also re-evaluated all methods under this standardized setting. As shown in Table 5, PiMT consistently outperforms all baselines, confirming that our improvements hold across different preprocessing configurations.

Table 5: F1-scores on SEED and BCI Competition IV 2b under the standardized EEGConformer preprocessing setting.

| Method | SEED | BCI Competition IV 2b |
|---|---|---|
| PatchTST | 0.945 | 0.826 |
| Bidirectional-Mamba | 0.956 | 0.823 |
| EEGConformer | 0.955 | 0.855 |
| **PiMT** | **0.971** | **0.869** |

## G  HYPER-PARAMETER TUNING

Our implementation consists of two primary stages: representation learning and task-specific fine-tuning. During the representation learning stage, we pre-trained the model using mask and reconstruction objectives to learn robust representations transferable across various downstream tasks. The representation model is trained on the entire 50 hours of free-living data, after which it is fine-tuned on each specific task before final evaluation on the corresponding test set.

The weighting coefficients ($\lambda$) for the pre-training objectives were selected heuristically based on empirical observations. We initialized all $\lambda$ values to 1 and monitored the convergence dynamics of individual loss terms. We found that the autoencoding loss ($\mathcal{L}_{AE}$) and masked reconstruction loss ($\mathcal{L}_{MR}$) converged more slowly than others; their weights were therefore increased to 2 to encourage balanced training. While we did not conduct a full hyperparameter sweep, this adjustment yielded more stable convergence without introducing instability.

To further optimize performance, we performed a grid search over key hyperparameters. Throughout the experiment, we used AdamW optimizer with 0.01 weight decay. During the pre-training stage, the batch size was fixed at 256, and the learning rate was scheduled from 0.01 to 0.001 using a cosine decay scheduler. For the backbone architecture, we adopted a Bidirectional Mamba model with 8 layers and a hidden dimension of 16. The masking ratio for the pre-training was fixed at 50%. During the fine-tuning stage, the batch size was fixed for all tasks, 10 for Gaze and 8 for the remaining tasks. The learning rate followed a cosine decay schedule from 0.001 to 0.00001. All experiments were repeated 3 times and the results are reported as the mean and standard deviation. All models were implemented using PyTorch, and the experimental evaluations were conducted on NVIDIA A100-SXM-80GB GPUs.

## H  BACKBONE COMPARISON: MAMBA VS. TRANSFORMER

We adopt Bidirectional-Mamba (Zhu et al., 2024) as our backbone architecture, which has demonstrated state-of-the-art performance across various time-series tasks (Zerveas et al., 2021; Song et al., 2022). In addition to Mamba, Transformer-based architectures are also widely used as backbones in time-series foundation models. In our main evaluation (Section 5.2), PatchTST (Nie et al., 2023) showed strong performance; notably, it represents a Transformer-based architecture without incorporating PiMT tokenization. We have shown the effectiveness of PiMT tokenization in Section 5.4. To further demonstrate the effectiveness our system design, we replace the Bidirectional-Mamba with a Transformer backbone for a fair comparison. Specifically, we conduct the experiments with and without pre-training on the free-living dataset.

As shown in Figure 10, the Mamba-based model consistently outperformed the Transformer-based model under all settings, achieving a 6.4% improvement without pre-training and an 8.5% gain with pre-training. These results confirm that Mamba is a strong architectural choice for ExG signal modeling.

## I  IMPACT OF PRE-TRAINING

To isolate the effect of the reconstruction-based pre-training from the contribution of PiMT, we evaluated model performance without applying PiMT. We compared (i) Bidirectional-Mamba and (ii) a Transformer-based architecture (PatchTST), each trained with and without pre-training under

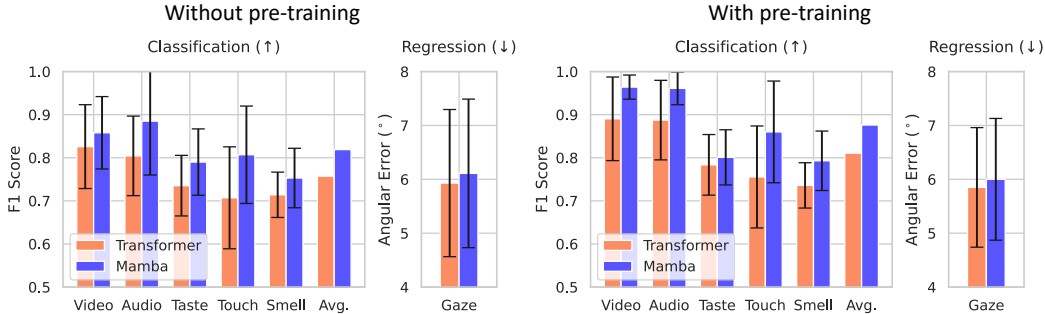

Figure 10: Comparison of Mamba and Transformer backbones.

Table 6: Performance of Transformer and Bidirectional-Mamba models with and without pre-training, excluding PiMT. Classification results are in F1-score, and gaze tracking performance is in angular error.

| | Classification (↑) | | | | | | Regression (↓) |
|---|---|---|---|---|---|---|---|
| Method | Video | Audio | Taste | Touch | Smell | Avg. | Gaze |
| *Without pre-training* | | | | | | | |
| Transformer | $0.771_{\pm 0.146}$ | $0.749_{\pm 0.113}$ | $0.731_{\pm 0.092}$ | $0.686_{\pm 0.119}$ | $0.681_{\pm 0.049}$ | 0.724 | $6.47°_{\pm 1.28°}$ |
| Bidirectional-Mamba | $0.820_{\pm 0.102}$ | $0.858_{\pm 0.113}$ | $0.733_{\pm 0.060}$ | $0.762_{\pm 0.101}$ | $0.722_{\pm 0.067}$ | 0.779 | $6.53°_{\pm 1.16°}$ |
| *With pre-training* | | | | | | | |
| Transformer | $0.807_{\pm 0.146}$ | $0.786_{\pm 0.147}$ | $0.697_{\pm 0.099}$ | $0.700_{\pm 0.131}$ | $0.670_{\pm 0.082}$ | 0.732 | $6.42°_{\pm 1.34°}$ |
| Bidirectional-Mamba | $0.891_{\pm 0.067}$ | $0.901_{\pm 0.084}$ | $0.741_{\pm 0.106}$ | $0.806_{\pm 0.118}$ | $0.726_{\pm 0.092}$ | 0.813 | $6.23°_{\pm 1.13°}$ |

identical settings. All models receive the same minimally processed input and share comparable architectural configurations.

Table 6 shows the results. Pre-training consistently improves performance across both architectures. For Bidirectional-Mamba, pre-training increases the average classification F1-score from 0.779 to 0.813 and reduces gaze tracking error from 6.53 to 6.23. Similarly, for the Transformer architecture, pre-training improves the average F1-score from 0.724 to 0.732 and slightly reduces gaze error from 6.47 to 6.42. These results demonstrate that the proposed pre-training strategy provides measurable benefits even without PiMT, indicating that the learned representations from free-living data are intrinsically useful.

## J EFFECT OF PRE-TRAINING COMPONENTS

Our pre-training framework on free-living data comprises six reconstruction-based tasks designed for unlabeled ExG signals: Autoencoding (AE), Masked Reconstruction (MR), (frequency) Amplitude Reconstruction (A), (frequency) Phase Reconstruction (P), Masked Amplitude Reconstruction (MA), and Masked Phase Reconstruction (MP). We assessed the contribution of each component by performing ablation experiments.

Table 7 depicts the results. Although most ablation settings still achieve relatively strong performance, highlighting the overall effectiveness of pre-training, all are consistently lower than the full combination (0.876), confirming the benefit of jointly using all reconstruction tasks. The performance drops in individual ablations are modest, as complementary tasks help maintain efficacy. However, we observed task-specific sensitivities: for instance, MR, A, MA, and MP are particularly important for gaze tracking, where excluding them led to a notable increase in error. Meanwhile, AE, MR, and phase-related reconstructions strongly influence taste classification, where their removal caused meaningful performance degradation. These findings suggest that temporal- versus frequency-focused reconstruction tasks contribute differently depending on the task, reflecting the distinct feature requirements of each modality.

Table 7: Pre-training ablation. Classification results are in F1-score, and gaze tracking performance is in angular error.

| Method | Classification (↑) | | | | | | Regression (↓) |
| | Video | Audio | Taste | Touch | Smell | Avg. | Gaze |
| --- | --- | --- | --- | --- | --- | --- | --- |
| PiMT (Ours) | $0.964 \pm 0.028$ | $0.961 \pm 0.038$ | $0.801 \pm 0.064$ | $0.860 \pm 0.118$ | $0.793 \pm 0.069$ | 0.876 | $6.00° \pm 1.13°$ |
| w/o AE | $0.970 \pm 0.026$ | $0.959 \pm 0.042$ | $0.806 \pm 0.066$ | $0.852 \pm 0.125$ | $0.778 \pm 0.085$ | 0.873 | $6.00° \pm 1.09°$ |
| w/o MR | $0.962 \pm 0.030$ | $0.956 \pm 0.043$ | $0.798 \pm 0.058$ | $0.859 \pm 0.122$ | $0.783 \pm 0.082$ | 0.872 | $6.10° \pm 1.09°$ |
| w/o A | $0.967 \pm 0.028$ | $0.955 \pm 0.037$ | $0.816 \pm 0.063$ | $0.850 \pm 0.119$ | $0.768 \pm 0.079$ | 0.871 | $6.19° \pm 1.23°$ |
| w/o MA | $0.965 \pm 0.032$ | $0.960 \pm 0.040$ | $0.818 \pm 0.062$ | $0.849 \pm 0.124$ | $0.774 \pm 0.089$ | 0.873 | $6.12° \pm 1.20°$ |
| w/o P | $0.979 \pm 0.020$ | $0.961 \pm 0.041$ | $0.803 \pm 0.061$ | $0.856 \pm 0.127$ | $0.764 \pm 0.086$ | 0.873 | $5.98° \pm 1.15°$ |
| w/o MP | $0.971 \pm 0.030$ | $0.960 \pm 0.040$ | $0.798 \pm 0.062$ | $0.857 \pm 0.120$ | $0.777 \pm 0.082$ | 0.873 | $6.15° \pm 1.26°$ |

## K  COMPARISON WITH DIFFERENT TOKENIZATION METHODS

To evaluate the effectiveness of PiMT, we compare it with two widely used time–frequency tokenization approaches: Short-Time Fourier Transform (STFT), Continuous Wavelet Transform (CWT), and raw signal. All methods are evaluated on DailySense using the same Bidirectional-Mamba backbone and identical training settings to ensure fair comparison.

**STFT.** We generate frequency-domain features by sliding a fixed window over time. The frequency vector extracted from each window is treated as a token and passed into the same learnable tokenizer used in our framework.

**CWT.** We compute continuous time–frequency coefficients using the Morlet wavelet, defining 12 frequency scales between 0.5 and 75 Hz for fairness. The resulting time–frequency representation preserves timestamps without additional segmentation. We then apply our patching scheme, and the frequency–time features within each patch are used as a single token.

**Raw Signal.** As a baseline, we also evaluate direct patch-based tokenization on the raw signal without explicit frequency decomposition.

Table 8: Comparison of different tokenization strategies on the DailySense dataset. We report classification F1 scores (↑) and gaze regression error in degrees (↓).

| Tokenization | Classification (↑) | | | | | | Regression (↓) |
| | Video | Audio | Taste | Touch | Smell | Avg. | Gaze |
| --- | --- | --- | --- | --- | --- | --- | --- |
| Raw Signal | $0.820 \pm 0.101$ | $0.858 \pm 0.113$ | $0.733 \pm 0.060$ | $0.762 \pm 0.100$ | $0.722 \pm 0.067$ | 0.779 | $6.53° \pm 1.16°$ |
| STFT | $0.800 \pm 0.101$ | $0.855 \pm 0.112$ | $0.731 \pm 0.066$ | $0.736 \pm 0.101$ | $0.721 \pm 0.047$ | 0.769 | $9.04° \pm 0.44°$ |
| CWT | $0.863 \pm 0.105$ | $0.881 \pm 0.109$ | $0.765 \pm 0.047$ | $0.800 \pm 0.096$ | $0.757 \pm 0.064$ | 0.813 | $6.57° \pm 1.34°$ |
| **PiMT** | $\mathbf{0.858} \pm \mathbf{0.084}$ | $\mathbf{0.885} \pm \mathbf{0.125}$ | $\mathbf{0.790} \pm \mathbf{0.077}$ | $\mathbf{0.807} \pm \mathbf{0.113}$ | $\mathbf{0.753} \pm \mathbf{0.069}$ | **0.819** | $\mathbf{6.11°} \pm \mathbf{1.20°}$ |

Table 8 shows the results. Frequency-domain approaches (STFT and CWT) provide meaningful representations for certain tasks, particularly Video and Smell, suggesting that explicit spectral structure can be beneficial. However, STFT exhibits degraded gaze performance (9.04°), likely due to loss of fine-grained temporal information. CWT improves over STFT but remains below PiMT in overall performance.

PiMT achieves the best average F1-score (0.819) and the lowest gaze error (6.11°), demonstrating strong performance across all tasks. By decomposing the signal into multiple physiology-informed bands while preserving time-domain structure within each band, PiMT balances frequency awareness with temporal fidelity, which is particularly important for temporally sensitive tasks such as gaze tracking.

## L  IMPACT OF PATCH SIZE

We discuss the impact of different patch sizes. Specifically, we selected the patch size empirically based on performance trends across tasks. A sensitivity study illustrating the effect of different patch sizes is presented in Table 9. A smaller patch size provides less contextual information for each

Table 9: Ablation study on the impact of patch size. We report classification F1 scores (↑) and gaze regression error in degrees (↓). Our method (0.5 sec patch size) achieves the best overall balance across tasks.

| Patchsize | Classification (↑) | | | | | | Regression (↓) |
|---|---|---|---|---|---|---|---|
| | Video | Audio | Taste | Touch | Smell | Avg. | Gaze |
| 0.25 sec | $0.977 \pm 0.020$ | $0.967 \pm 0.036$ | $0.776 \pm 0.065$ | $0.849 \pm 0.124$ | $0.741 \pm 0.098$ | 0.862 | $6.23° \pm 1.26°$ |
| **0.5 sec (Ours)** | $\mathbf{0.964} \pm \mathbf{0.028}$ | $\mathbf{0.961} \pm \mathbf{0.038}$ | $\mathbf{0.801} \pm \mathbf{0.064}$ | $\mathbf{0.860} \pm \mathbf{0.118}$ | $\mathbf{0.793} \pm \mathbf{0.069}$ | **0.876** | $\mathbf{6.00°} \pm \mathbf{1.13°}$ |
| 1.0 sec | $0.962 \pm 0.034$ | $0.945 \pm 0.054$ | $0.821 \pm 0.063$ | $0.836 \pm 0.130$ | $0.784 \pm 0.088$ | 0.870 | $6.10° \pm 1.09°$ |
| 2.0 sec | $0.947 \pm 0.039$ | $0.932 \pm 0.074$ | $0.776 \pm 0.101$ | $0.823 \pm 0.126$ | $0.769 \pm 0.105$ | 0.850 | $6.24° \pm 1.05°$ |

classification window, which may limit performance. However, it benefits gaze regression, as the participant's gaze is more likely to remain fixed within a shorter temporal window. In contrast, larger patch sizes offer more temporal context for classification tasks but increase the likelihood of gaze shifts or overlapping signals from multiple classes, potentially degrading both classification and gaze estimation performance. We observed that a patch size of 0.5 seconds provides the best trade-off, yielding strong performance across both classification and regression tasks.

## M    EVALUATION WITH FROZEN ENCODERS

Table 10: Evaluation under frozen encoder settings. We report classification F1 scores (↑) and gaze regression error in degrees (↓). PiMT maintains strong performance even when the encoder is frozen.

| Method | Classification (↑) | | | | | | Regression (↓) |
|---|---|---|---|---|---|---|---|
| | Video | Audio | Taste | Touch | Smell | Avg. | Gaze |
| PiMT (finetune) | $0.964 \pm 0.028$ | $0.961 \pm 0.038$ | $0.801 \pm 0.064$ | $0.860 \pm 0.118$ | $0.793 \pm 0.069$ | 0.876 | $6.00° \pm 1.13°$ |
| PatchTST (frozen) | $0.812 \pm 0.146$ | $0.736 \pm 0.148$ | $0.717 \pm 0.121$ | $0.696 \pm 0.131$ | $0.666 \pm 0.091$ | 0.725 | $7.23° \pm 1.19°$ |
| PiMT (frozen) | $0.957 \pm 0.031$ | $0.944 \pm 0.050$ | $0.799 \pm 0.060$ | $0.863 \pm 0.105$ | $0.771 \pm 0.073$ | 0.866 | $6.36° \pm 1.02°$ |

To further evaluate the intrinsic quality of the learned representations, we assess model performance under a frozen-encoder setting. After pre-training, the encoder is kept fixed, and only lightweight task-specific decoder layers are trained for each downstream task. This protocol isolates the contribution of the pre-trained representation and reflects the feasibility of parameter-efficient adaptation compared to full end-to-end fine-tuning.

Table 10 reports the results. PiMT with full fine-tuning achieves an average F1-score of 0.876 and a gaze error of 6.00. When freezing the encoder, PiMT maintains strong performance (0.866 average F1, 6.36 gaze error), indicating that the pre-trained features generalize effectively across diverse tasks even without encoder updates. For comparison, PatchTST under the frozen setting achieves an average F1-score of 0.725 and a gaze error of 7.23, consistently below its fully fine-tuned counterpart. Notably, the gaze task exhibits a larger performance drop under frozen settings, suggesting that temporally sensitive tasks benefit more from encoder-level adaptation.

## N    EFFICIENCY ANALYSIS

Table 11: Runtime performance of PiMT on smartphone (Samsung Galaxy S24).

| Metric | Value |
|---|---|
| Inference Latency | 25 ms |
| Memory Usage | 266 MB (3.6%) |
| CPU Usage | 20.3% |
| Model Size (ONNX) | 2.0 MB |

We evaluated the runtime overhead of our method on a commercial smartphone (Samsung Galaxy S24), which serves as a practical companion device for earphone-based systems. Since NeuroBuds

Table 12: Performance of PiMT compared to PatchTST on the DailySense dataset under three data split settings: within-session, cross-session, and cross-subject.

| Method | Classification (↑) | | | | | | Regression (↓) |
|---|---|---|---|---|---|---|---|
| | Video | Audio | Taste | Touch | Smell | Avg. | Gaze |
| *Within-session* | | | | | | | |
| PatchTST | $0.807 \pm 0.146$ | $0.786 \pm 0.146$ | $0.697 \pm 0.099$ | $0.700 \pm 0.131$ | $0.670 \pm 0.082$ | 0.732 | $6.42° \pm 1.33°$ |
| PiMT (Ours) | $0.964 \pm 0.028$ | $0.961 \pm 0.038$ | $0.801 \pm 0.064$ | $0.860 \pm 0.118$ | $0.793 \pm 0.069$ | 0.876 | $6.00° \pm 1.13°$ |
| *Cross-subject* | | | | | | | |
| PatchTST | $0.654 \pm 0.047$ | $0.595 \pm 0.064$ | $0.561 \pm 0.047$ | $0.553 \pm 0.064$ | $0.539 \pm 0.052$ | 0.580 | $7.07° \pm 1.25°$ |
| PiMT (Ours) | $0.612 \pm 0.088$ | $0.578 \pm 0.082$ | $0.593 \pm 0.038$ | $0.577 \pm 0.092$ | $0.571 \pm 0.035$ | 0.586 | $7.78° \pm 0.95°$ |
| *Cross-session* | | | | | | | |
| PatchTST | $0.658 \pm 0.202$ | $0.656 \pm 0.157$ | $0.695 \pm 0.101$ | $0.639 \pm 0.097$ | $0.611 \pm 0.068$ | 0.652 | $7.56° \pm 1.33°$ |
| PiMT (Ours) | $0.697 \pm 0.249$ | $0.698 \pm 0.188$ | $0.704 \pm 0.106$ | $0.763 \pm 0.156$ | $0.639 \pm 0.146$ | 0.700 | $6.98° \pm 1.51°$ |

supports real-time data streaming via BLE, we consider a deployment scenario where inference is offloaded to the smartphone.

Because the Mamba architecture is not yet supported on Android and lacks corresponding hardware acceleration, we substituted Mamba with a Transformer of *equivalent architecture and parameter size* (*e.g.*, number of layers, $d_{\text{model}}$). Prior work has shown that Transformers generally incur higher inference costs under comparable hardware acceleration (Gu & Dao, 2024). To preserve the core algorithmic behavior of PiMT, we retained PiMT and the 3D positional embeddings.

The resulting models were exported to ONNX and evaluated using 4-second input sliding windows (200 Hz sampling, with the same preprocessing as in the main experiments). The measured runtime performance is summarized in Table 11.

Overall, the results indicate that inference can be executed in real time at up to 40 Hz with minimal resource consumption. Preprocessing operations such as filtering and windowing can be performed directly on the NeuroBuds board. In addition, given that Mamba has been reported to offer $5\times$ higher throughput than Transformers, we anticipate supporting on-device inference with even lower overhead.

## O    GENERALIZATION TO UNSEEN SESSIONS AND USERS

We also tested generalization to unseen conditions during testing in *cross-subject* and *cross-session* scenarios. Cross-subject involves training and testing on different users, while cross-session assumes the model is tested on a different session from the same user, introducing a temporal shift. These domain shifts are open challenges for ExG-based tasks, with prior work (Fan et al., 2024) reporting over a 30% drop in accuracy. As shown in Table 12, our method also experienced a performance drop under the cross-subject setting (58.6%). In the cross-session setting, NeuroBuds showed stronger robustness, achieving 70.0% compared to PatchTST's 65.2%. Generalization to unseen conditions remains an open challenge and is a focus of our future research. Nevertheless, we believe that large-scale data collection enabled by the daily usability of NeuroBuds can play a key role in improving robustness in real-world applications.

## P    STATISTICAL SIGNIFICANCE

To assess whether our method provides statistically significant improvements over the baselines, we conduct paired Wilcoxon signed-rank tests on DailySense. Each task contains 6–9 participants, and for every participant we train each model with three random seeds. For a given seed, all models share the exact same train–test split; because the split strongly influences performance, constructing pairs at the seed level ensures a fair and properly controlled comparison. For each model pair, we therefore form paired samples based on participant–seed combinations (*e.g.*, 7 participants × 3 seeds = 21 paired samples), and the Wilcoxon test is applied to this aggregated set of paired differences. For the classification tasks, we test whether the performance differences are consistently positive (higher F1 is better), and for gaze estimation we test whether the differences are consistently

Table 13: Paired Wilcoxon $p$-values when comparing PiMT to each baseline on DailySense. Lower values indicate stronger evidence that PiMT outperforms the baseline.

| Method | Classification (↑) | | | | | | Regression (↓) |
|---|---|---|---|---|---|---|---|
| | Video | Audio | Taste | Touch | Smell | Avg. | Gaze |
| *Without pre-training* | | | | | | | |
| SVM | 4.77 e-07 *** | 4.77 e-07 *** | 7.63 e-06 *** | 7.45 e-09 *** | 4.77 e-07 *** | 9.61 e-20 *** | 4.44 e-05 *** |
| EEGNet | 1.64 e-04 *** | 1.21 e-04 *** | 1.23 e-02 * | 1.49 e-08 *** | 1.59 e-03 ** | 8.25 e-15 *** | 1.01 e-03 ** |
| DeepConvNet | 9.82 e-05 *** | 4.77 e-07 *** | 1.64 e-04 *** | 1.42 e-07 *** | 4.77 e-06 *** | 4.48 e-18 *** | 1.03 e-07 *** |
| TST | 1.17 e-04 *** | 1.19 e-05 *** | 1.92 e-02 * | 4.10 e-07 *** | 6.53 e-05 *** | 1.22 e-15 *** | 8.04 e-04 *** |
| PatchTST | 4.32 e-04 *** | 1.09 e-04 *** | 3.00 e-02 * | 1.49 e-08 *** | 1.65 e-03 ** | 2.65 e-14 *** | 2.14 e-03 ** |
| EEGConformer | 9.82 e-05 *** | 5.25 e-05 *** | 5.23 e-04 *** | 1.42 e-07 *** | 1.24 e-03 ** | 3.28 e-16 *** | 1.38 e-03 ** |
| Bidirectional-Mamba | 6.14 e-03 ** | 1.83 e-02 * | 1.56 e-02 * | 1.12 e-03 ** | 2.30 e-02 * | 1.01 e-07 *** | 1.15 e-07 *** |
| **PiMT (Ours)** | – | – | – | – | – | – | – |
| *With pre-training* | | | | | | | |
| PatchTST | 1.17 e-04 *** | 9.54 e-07 *** | 1.26 e-04 *** | 7.45 e-08 *** | 9.54 e-07 *** | 1.91 e-18 *** | 3.29 e-03 ** |
| **PiMT (Ours)** | – | – | – | – | – | – | – |

\* $p \leq 0.05$, \*\* $p \leq 0.01$, \*\*\* $p \leq 0.001$. Entries marked "–" correspond to self-comparisons.

Table 14: Performance on DailySense using different pre-training sources, showing our 50-hr free-living dataset outperforms larger controlled datasets.

| Pre-Training Dataset | Classification (↑) | | | | | | Regression (↓) |
|---|---|---|---|---|---|---|---|
| | Video | Audio | Taste | Touch | Smell | Avg. | Gaze |
| No PT | $0.858 \pm 0.084$ | $0.885 \pm 0.125$ | $0.790 \pm 0.077$ | $0.807 \pm 0.113$ | $0.753 \pm 0.069$ | 0.819 | $6.11° \pm 1.20°$ |
| TUAR (98.6 hrs) | $0.964 \pm 0.035$ | $0.950 \pm 0.049$ | $0.791 \pm 0.065$ | $0.824 \pm 0.124$ | $0.736 \pm 0.092$ | 0.853 | $5.95° \pm 1.17°$ |
| TUAR + TUSZ (498.6 hrs) | $0.964 \pm 0.024$ | $0.950 \pm 0.044$ | $0.803 \pm 0.076$ | $0.833 \pm 0.109$ | $0.741 \pm 0.094$ | 0.858 | $6.03° \pm 1.14°$ |
| **DailySense (Ours, 50 hrs)** | $\mathbf{0.964} \pm \mathbf{0.028}$ | $\mathbf{0.961} \pm \mathbf{0.038}$ | $\mathbf{0.801} \pm \mathbf{0.064}$ | $\mathbf{0.860} \pm \mathbf{0.118}$ | $\mathbf{0.793} \pm \mathbf{0.069}$ | **0.876** | $\mathbf{6.00°} \pm \mathbf{1.13°}$ |

negative (lower angular error is better). For the "Avg." column, we pool paired differences across all five classification tasks before applying the test. As shown in Table 13, PiMT achieves statistically significant improvements over nearly all baselines and modalities, often with extremely small $p$-values, demonstrating that the gains are consistent across participants and robust to seed-level variation.

## Q   PRE-TRAINING USING PUBLIC CONTROLLED EXG DATASETS

We further examine how our 50-hour free-living DailySense dataset compares to pre-training on larger publicly available ExG corpora collected in controlled settings. Specifically, we evaluate models pre-trained on TUAR (Hamid et al., 2020) and TUSZ (Shah et al., 2018), two widely used pre-training datasets in recent EEG foundation models (Jiang et al., 2024; Cui et al., 2024; Fang et al., 2025). TUAR, a curated subset of TUEG (Obeid & Picone, 2016), contains annotations for five artifact types—including eye movements, chewing, and muscle activity—making it relevant to our downstream tasks such as gaze tracking. TUSZ provides extensive seizure annotations and is among the largest publicly available EEG corpora. Since these datasets use different electrode montages, we select electrodes with closest spatial matches—F7, F8, T3, T4, T5, T6, O1, and O2 in the 10–20 system—for pre-training. We consider two pre-training configurations: (1) TUAR alone (98.6 hours) and (2) TUAR combined with a subset of TUSZ (498.6 hours total), representing moderate- and large-scale controlled EEG datasets, respectively. As shown in Table 14, pre-training on DailySense achieves the best average performance across all five classification tasks and yields competitive gaze estimation accuracy, despite its substantially smaller size. This highlights the power of learning more generalizable and robust representations from free-living data, which better capture the natural variability present in real-world human behavior than controlled laboratory recordings. Given the ease of free-living data collection, DailySense can be scaled more readily, which would further strengthen these advantages.

