# OpenReview forum: "Beyond Hearing: Learning Task-Agnostic ExG Representations from Earphones via Physiology-Informed Tokenization"
_ICLR.cc/2026/Conference — ICLR 2026 Poster_

### Official Review · Reviewer_ZFuK · 2025-10-18

**Soundness:** 4
**Presentation:** 3
**Contribution:** 3
**Rating:** 6
**Confidence:** 4

**Summary:**

The author introduces a new design of earphone based wearable device incorporating several electrode sensors to capture EEG, EOG, and EMG. A new benchmark dataset DailySense, containing 50 hours of data from 22 participants, is proposed. A paradigm pre-trained model is presented and indicates leading performance across several downstream human sensing tasks against baselines.

**Strengths:**

- From the hardware design perspective, the design of the proposed wearable is ingenious. The earbuds based device is able to capture brain activity, heart activity, and facial muscle activity simultaneously. The device is lightweight and low-cost.
- From the modeling perspective, the proposed Physiology informed Multi-band Tokenization (PiFT) is novel. The design of this tokenization process is reasonable in handling the sensing data with multiple channels and varied frequency-band of interest.
- The experimental results are comprehensive, where varied baselines are included in comparison. Ablation studies are conducted to justify the efficiency of the proposed components.

**Weaknesses:**

## The modeling approach requires some additional justification.
- If the emphasis is on the proposed PiFT mechanism, then it would be better to show that the gain in performance is consistent and independent of the backbone. The experimental setting could either be:
    - 1) With PiFT fixed, other than just modeling with Bidirectional Mamba, also model with other backbone such as Transformers.
    - 2) With the same backbone, compare PiFT performance against other different tokenization mechanisms, for example short term Fourier transform and continuous wavelet transform which are widely used in signal processing applications.
- Recognizing the relatively higher uncertainty (due to more hyper-parameters and configuration) and timely cost on evaluate models’ representation with finetuning, I would suggest evaluate with linear probing, and compare against several recent and open-sourced (i.e. can be used off-the-shelf) models that are pretrained on wearable signals and the design of the model are intend to be channel agnostic, such as CBraMod [1] and NormWear [2].

## There are some limitations on the collected data.
Since collecting data often contain difficulties, given the thoughtful device design and reasonable modeling scheme, this issue is not a major concern, but it still deserves to be mentioned.
- The size of the collected data is a bit limited, where 50 hours of wearable signals from 22 participants is considered a small dataset in the wearable sensing domain. Also the demographic diversity of the dataset is also a limitation.
- The downstream tasks constructed from the collected dataset tend to be relatively straightforward, with the majority formulated as binary classification problems.
- A well-rounded data sheet reporting the statistics of the collected data and the 4 public datasets will improve the presentation a lot. For example, the total hours of data, participants information, and distribution of different sensor signals, etc.

## Reference
[1] Wang, Jiquan, et al. "Cbramod: A criss-cross brain foundation model for eeg decoding." 2024.

[2] Luo, Yunfei, et al. "Toward Foundation Model for Multivariate Wearable Sensing of Physiological Signals." 2024.

**Questions:**

Most of my suggestions are comprised in the weaknesses above. Overall the presentation of the paper is clear and all the notions are explained comprehensively.

---

> ### Author Response · Authors · 2025-11-23
> **Response to Reviewer ZFuk (1/2)**
>
> Dear Reviewer ZFuK,
>
> We truly appreciate your insightful suggestions. We are encouraged by the reviewers’ recognition of our **solid prototype and free-living dataset**, which together enable scalable ExG sensing in the wild (highlighted by mTe5, ejoF, ZFuK). We are also grateful for the positive assessment of our **PiMT and pre-training**, which reviewers found to be principled, novel, and broadly effective across tasks (ejoF, UWPo, ZFuK). Finally, we are heartened by the reviewers’ acknowledgement of our **comprehensive experiments**, including evaluations on both our new dataset and four public benchmarks, as well as the **ablation studies** that clarify the contribution of each component (mTe5, ejoF, UWPo, ZFuK).
>
> We have carefully addressed your feedback and questions below.
>
> ---
>
> **[W1-1] Justification of PiMT**
>
> We appreciate your suggestions regarding backbone independence and comparisons with different tokenizations. In response, we provide the following experiments.
>
> 1. PiMT + Transformer
>
> Appendix H has included an experiment replacing Bidirectional-Mamba with Transformer while keeping PiMT fixed. Moreover, we also provide a direct comparison between Transformer and PiMT + Transformer under both the pre-training (i.e., pt) and no-pre-training settings.
>
> | Setting | Video | Audio | Taste | Touch | Smell | Avg. | Gaze |
> |-|-|-|-|-|-|-|-|
> | Transformer (w/o pt) | 0.771 | 0.749 | 0.731 | 0.686 | 0.681 | 0.724 | 6.47 |
> | PiMT + Transformer (w/o pt) | 0.826 | 0.804 | 0.735 | 0.707 | 0.714 | 0.757 | 5.93 |
> | Transformer (w/ pt) | 0.807 | 0.786 | 0.697 | 0.700 | 0.670 | 0.732 | 6.42 |
> | PiMT + Transformer (w/ pt) | 0.890 | 0.887 | 0.784 | 0.756 | 0.736 | 0.811 | 5.85 |
>
> The results confirm that the gains are not specific to Bidirectional-Mamba.
>
> 2. PiMT vs. other tokenizations
>
> We compared PiMT with two widely used methods: Short-Time Fourier Transform (STFT) and Continuous Wavelet Transform (CWT). For STFT, we generated frequency features by sliding a fixed window over time. Each window’s frequency feature vector was used as a token and passed into the same learnable tokenizer as ours. For CWT, we extracted continuous time-frequency coefficients while preserving the timestamps without additional segmentation. To ensure fairness, we defined 12 frequency scales between 0.5 and 75 using the Morlet wavelet. Finally, we segmented the time using our patching scheme. The frequency-time domain features within each patch were used as a single token and fed into the tokenizer.
>
> | Tokenization | Video | Audio | Taste | Touch | Smell | Avg. | Gaze |
> |-|-|-|-|-|-|-|-|
> | raw signal | 0.820 | 0.858 | 0.733 | 0.762 | 0.722 | 0.779 | 6.53 |
> | STFT | 0.800 | 0.855 | 0.731 | 0.736 | 0.721 | 0.769 | 9.04 |
> | CWT | 0.863 | 0.881 | 0.765 | 0.800 | 0.757 | 0.813 | 6.57 |
> | PiMT | 0.858 | 0.885 | 0.790 | 0.807 | 0.753 | 0.819 | 6.11 |
>
> The table above shows the results on DailySense using the Bidirectional-Mamba backbone with different tokenizations. We observe that frequency-domain approaches provide meaningful features for tasks such as Video and Smell, while PiMT achieves the best overall performance. In particular, PiMT excels on temporally sensitive tasks such as Gaze, which depend on fine time-domain patterns that STFT/CWT discard. By preserving these temporal structures through multi-band time-domain filtering, PiMT achieves strong performance on all DailySense tasks.
>
> ---
>
> **[W1-2] Comparison of Learned Representations**
>
> Thank you for the constructive suggestion. We conducted extra experiments including (1) linear probing, where we freeze the encoder and training only a single linear layer, and (2) then compare PiMT against recent open-sourced channel-agnostic models CBraMod and NormWear across DailySense tasks. We followed their official implementations and used publicly released weights.
>
> | Method | Video | Audio | Taste | Touch | Smell | Avg. | Gaze |
> |-|-|-|-|-|-|-|-|
> | CBraMod | 0.733 | 0.665 | 0.568 | 0.655 | 0.592 | 0.642 | 14.831 |
> | NormWear | 0.795 | 0.759 | 0.663 | 0.824 | 0.736 | 0.707 | 9.00 |
> | PiMT (linear-prob) | 0.869 | 0.884 | 0.739 | 0.772 | 0.701 | 0.793 | 8.40 |
>
> With frozen encoders, PiMT achieves the strongest average performance across all tasks (Table below), demonstrating that its representations transfer more effectively than these recent open-sourced models. We attribute this to PiMT’s physiology-informed multi-band design and to pre-training on naturalistic, free-living data that better match the characteristics of our downstream tasks.
>
> To further validate this, we additionally compared PiMT pre-training against different lab-controlled large-scale EEG datasets. We included these analyses in Appendix N, along with our response to Reviewer mTe5 regarding dataset and ML design effectiveness.

---

> ### Author Response · Authors · 2025-11-23
> **Response to Reviewer ZFuk (2/2)**
>
> **[W2-1] Limited Hours of Collected Data**
>
> We appreciate the reviewer’s point and agree that collecting large-scale ExG data is challenging. Existing datasets are often constrained by in-lab protocols and typically contain shorter total recording durations (e.g., DREAMER ~23 hours, SEED ~45 hours). In contrast, our DailySense dataset includes 50 hours of diverse free-living signals and 20 hours of task-specific recordings, totaling 70 hours. We also provide the detailed data sheet below.
>
> A key advantage of our work is the low-cost, easy-to-wear earphone prototype, which substantially reduces the barrier to recruiting larger and more diverse cohorts. This design enables natural, long-duration free-living recordings and provides a realistic path toward scaling to larger and more diverse cohorts in future work. We will make this clearer in the revised manuscript.
>
> ---
>
> **[W2-2] Downstream Tasks Construction**
>
> As the first unified earphone-based system to explore whether minimally intrusive ear-ExG signals can reflect information for human senses, our dataset design strictly follows established brain-sensing protocols traditionally collected by EEG headsets [r1,r2,r3] which are often binary settings.
>
> We would like to note that our evaluation on existing benchmarks includes more complex tasks, for example, SEED is 3-class and Sleep-EDF is the 5-class sleep stage classification task. PiMT performs strongly in these settings, demonstrating that the model is not limited to binary tasks and generalizes to higher-complexity public datasets. In future work, we believe that our work can be a strong basis for exploring larger scale and sophisticated tasks.
>
> [r1] Amini, et al. "Surface roughness classification in dynamic touch using EEG signals." ICEE 2022.
>
> [r2] Eldeeb, et al. "EEG-based trial-by-trial texture classification during active touch." Scientific reports 2020.
>
> [r3] Libert, et al. "Predicting premature video skipping and viewer interest from EEG recordings." Entropy 2019.
>
> ---
>
> **[W2-3] Data Sheet**
>
> | Dataset | Device | Setting | Signals | Participants | Total Hours |
> |-|-|-|-|-|-|
> | DailySense | Earphone | Daily free-living | ExG | 22 | 70 |
> | DREAMER | Headset | Film watching | EEG | 23 | 23 |
> | SEED | Headset | Film watching | EEG | 15 | 45 |
> | Sleep-EDF | Polysomn. | Sleep lab | EEG, EOG, EMG | 15 | 45 |
> | BCI Competition IV 2b | Controlled | Controlled | EEG, EOG | 9 | 6 |

---

> > ### Comment · Reviewer_ZFuK · 2025-11-24
> > **Response to Author**
> >
> > I thank the authors for the thorough response. The additional experimental results greatly help contextualize the framework’s performance and behavior. These include:
> > - a. Ablation studies that justify the chosen modules, including evaluations of alternative backbones and initial embeddings.
> > - b. Further comparisons with existing off-the-shelf models, which strengthen the evidence for the effectiveness of the proposed methodology.
> >
> > All of my concerns have been addressed adequately. My only remaining suggestion is to consider integrating these additional experimental results into the revised version, as they would further enhance the clarity and completeness of the presentation. The paper has improved in both content and presentation, and I have raised my score accordingly.

---

> > > ### Author Response · Authors · 2025-11-25
> > > **Response to Reviewer ZFuk**
> > >
> > > Thank you very much for your thoughtful follow-up and for updating your score.
> > >
> > > We are delighted that the additional analyses helped fully address your concerns. We also appreciate your suggestion to integrate these results into the revised manuscript. We will incorporate the key findings to further improve clarity.
> > >
> > > Thank you again for the careful review and constructive guidance.

---

### Official Review · Reviewer_UWPo · 2025-10-21

**Soundness:** 3
**Presentation:** 4
**Contribution:** 3
**Rating:** 8
**Confidence:** 3

**Summary:**

This paper introduces pre-training methods and a new tokenization scheme for electrophysiological (ExG) signal modeling.  Their proposed pre-training methods and Physiology-informed Multi-band Tokenization scheme aim to address the challenge of ExG model generalization.  Evaluated on common ExG benchmark datasets, the introduced models outperform existing baselines.  The authors additionally collect the DailySense dataset, a free-living dataset of ExG data collected while participants perform tasks targeting different human senses, to further benchmark the generality of their proposed model.

**Strengths:**

- The authors tackle the important challenge of model generalization in ExG modeling.  In many cases in this modeling domain, generalization is largely accounted for with dataset scale, requiring more resources.  However, the novel tokenization scheme proposed in this work achieves improvements in generalization by developing an encoding scheme that seeks to explicitly account for known physiological principals of ExG signals.
- Proposed strategies (pre-training and PiMT) consistently outperform baseline models across a variety of common (and new) benchmarks.
- Ablation studies are provided to verify the relevance of different pre-training strategies, frequency bands, backbones, and other design choices.

**Weaknesses:**

- The physiologically informed aspect of PiMT assumes electrode configurations that can yield signals of interest at each frequency band and does not account for mixing of these signals, potentially limiting generality of the scheme to different hardware configurations.  For instance, computing EOG signal features from occipital electrodes may not have much physiological meaning (the model would still likely learn features relevant to the training task, but the physiological intuition diminishes).
- Baseline models evaluated in Table 1 are all evaluated on data collected from NeuroBuds.  These baselines may not be readily applicable to minimally pre-processed data and performance improvements may be observed when using data that has been processed further.  PiMT explicitly performs filtering across various frequency bands of interest, so implicitly includes additional preprocessing that other methods may not benefit from.

**Questions:**

- What is the performance of the model without PiMT but with a proposed pre-training strategy?  Can other model architectures benefit from a similar pre-training strategy? If I am understanding correctly, table 1 shows model performance with and without pre-training but always uses PiMT.
- The results of this work primarily tackle task-generalization.  Do you expect that this physiologically informed tokenization scheme can also help extract features that are beneficial for cross-subject generalization?
- After fine-tuning, how well do the learned features generalize to the evaluation tasks of interest when the encoder is completely frozen (as opposed to enabling task-specific model updates during fine-tuning)?

**Details Of Ethics Concerns:**

Data collection from human subjects was approved by IRB.

---

> ### Author Response · Authors · 2025-11-23
> **Response to Reviewer UWPo (1/2)**
>
> Dear Reviewer UWPo,
>
> We sincerely appreciate your thoughtful and constructive feedback. We are encouraged by the reviewers’ recognition of our **solid prototype and free-living dataset**, which together enable scalable ExG sensing in the wild (highlighted by mTe5, ejoF, ZFuK). We are also grateful for the positive assessment of our **PiMT and pre-training**, which reviewers found to be principled, novel, and broadly effective across tasks (ejoF, UWPo, ZFuK). Finally, we are heartened by the reviewers’ acknowledgement of our **comprehensive experiments**, including evaluations on both our new dataset and four public benchmarks, as well as the **ablation studies** that clarify the contribution of each component (mTe5, ejoF, UWPo, ZFuK).
>
> We have reviewed your comments and provide our responses below.
>
> ---
>
> **[W1] Generality of PiMT to Different Hardware Configurations**
>
> Thank you for the thoughtful observation. We agree that different hardware configurations may not provide clean physiological separation for every predefined band-electrode combination. PiMT is therefore designed to be *flexible*: we assume no prior knowledge about which electrodes or bands are informative for a given task, and use a physiology-inspired superset of frequency bands that allows the model to learn whichever components are useful in a data-driven manner. In your example, where EOG may be weak at a given electrode, we expect the model to naturally rely on more informative features (e.g., EEG activity) within the multi-band structure.
>
> We also agree that depending on the hardware, certain bands may contribute less, which suggests a meaningful direction for future refinement such as adaptive band optimization for token reduction. Finally, near-ear electrodes inherently capture mixed EEG/EOG/EMG signals, making such flexibility essential for real-world wearable hardware. We will clarify this intent in our manuscript.
>
> ---
>
> **[W2] Preprocessing of Baselines**
>
> We appreciate your concern. While we applied standard processing (notch, lowpass, and highpass filterings), it is possible that some baseline models may achieve higher performance when paired with additional preprocessing. For example, for gaze tracking, applying additional low-frequency filtering (e.g., 0.1–20 Hz) can enhance EOG-related components, and for tasks that rely heavily on frequency structure, generating explicit frequency-domain features (e.g., via FFT or CWT) may offer advantages.
>
> However, our goal is task-agnostic modeling. To ensure a fair and uniform comparison, all baseline models and PiMT receive the same minimally processed input. Specifically, introducing additional preprocessing (e.g., EOG-focused filtering or explicit frequency transforms) would implicitly encode assumptions that do not generalize across tasks, and thus we adopt a minimal preprocessing pipeline.
>
> To further validate fairness, we additionally evaluated models under different processing tokenization using CWT and STFT (presented in ZFuk [W1-1]). These analyses show that PiMT continues to outperform those alternatives across tasks. We will clarify these points and reference the additional comparisons in the revised manuscript.

---

> ### Author Response · Authors · 2025-11-23
> **Response to Reviewer UWPo (2/2)**
>
> **[Q1] Performance without PiMT**
>
> The 1-band baseline in Figure 4 corresponds to the same architecture and pre-training as ours, but without PiMT. Following your recommendation, we include a comparison that isolates the effect of PiMT by evaluating 1) Bidirectional-Mamba with and without pre-training, and 2) a Transformer-based architecture (PatchTST) under the same settings.
>
> | Setting | Video | Audio | Taste | Touch | Smell | Avg. | Gaze |
> |-|-|-|-|-|-|-|-|
> | Bidirectional-Mamba (w/o pt) | 0.820 | 0.858 | 0.733 | 0.762 | 0.722 | 0.779 | 6.53 |
> | Bidirectional-Mamba (w/ pt) | 0.891 | 0.901 | 0.741 | 0.806 | 0.726 | 0.813 | 6.23 |
> | Transformer (w/o pt) | 0.771 | 0.749 | 0.731 | 0.686 | 0.681 | 0.724 | 6.47 |
> | Transformer (w/ pt) | 0.807 | 0.786 | 0.697 | 0.700 | 0.670 | 0.732 | 6.42 |
>
> Pre-training clearly improves performance across both architectures even without PiMT, demonstrating the benefit of our pre-training approach. However, the main results table shows that the gains from PiMT remain substantially larger. This highlights that PiMT contributes not only to downstream performance but also to extracting richer representations during pre-training on free-living data. We will incorporate this discussion into the revised manuscript.
>
> ---
>
> **[Q2] PiMT for Cross-subject Generalization**
>
> We appreciate your insight. At the representation level, our results in Figure 7 indicate that downstream performance of pre-training PiMT excluding the target user’s free-living data is comparable to full pre-training with target user’s data, which highlights the model’s capability of learning generalizable representation to unseen users.
>
> However, cross-subject testing (i.e., fine-tuning on one subject and testing on another) remains an open challenge in ExG research due to substantial inter-subject variability (e.g., individual physiology, sensor placement). We view this broader problem as important ongoing work rather than something fully addressed by our current study. PiMT may provide a flexible representation that captures components that are stable across individuals. However, validating this potential will require larger and more diverse datasets, which we consider a direction for future work. We will clarify this scope in the revised manuscript.
>
> ---
>
> **[Q3] Frozen Encoders**
>
> Thank you for the insightful question. To evaluate our pre-trained encoder without task-specific tuning we froze the encoder after pre-training, and only the decoder layers were trained for each task. This setting reflects the quality of the pre-trained representation and also indicates the feasibility of lightweight fine-tuning compared to full end-to-end updates.
>
> | Method | Video | Audio | Taste | Touch | Smell | Avg. | Gaze |
> |-|-|-|-|-|-|-|-|
> | PiMT (finetune) | 0.964 | 0.961 | 0.801 | 0.860 | 0.793 | 0.876 | 6.00 |
> | PatchTST (frozen)  | 0.812 | 0.736 | 0.717 | 0.696 | 0.666 | 0.725 | 7.23 |
> | PiMT (frozen) | 0.957 | 0.944 | 0.799 | 0.863 | 0.771 | 0.866 | 6.36 |
>
> The results show that our pre-trained features generalize effectively across diverse tasks, even when the encoder is frozen. PatchTST yields viable performance, but consistently below its full fine-tuning results. Notably, the Gaze task exhibits a substantial drop, indicating that temporally sensitive tasks require encoder-level adaptation using sequential updates. For ours, the performance is slightly lower than the full fine-tuning (0.876), but the frozen encoder still provides strong representations.

---

> ### Comment · Reviewer_UWPo · 2025-11-25
> **Response to author's rebuttal**
>
> I appreciate the time that the authors have taken to address the weaknesses and questions that I raised in my initial review. These follow-up experiments provide additional evidence for the utility of PiMT and I encourage the authors to include the details of these experiments and relevant commentary into their paper.  I believe my original rating was appropriate (8: accept).

---

> > ### Author Response · Authors · 2025-11-26
> >
> > Thank you very much for your follow-up and for revisiting our work with the additional experiments. We are glad that the new analyses helped clarify the utility of PiMT and addressed your earlier questions. We will incorporate these results into the revised manuscript to further strengthen the clarity of our work.
> > Thank you again for your constructive feedback and for maintaining your positive assessment.

---

### Official Review · Reviewer_ejoF · 2025-10-29

**Soundness:** 3
**Presentation:** 3
**Contribution:** 3
**Rating:** 6
**Confidence:** 3

**Summary:**

This paper introduces a new framework for scalable, task-agnostic Electrophysiological (ExG) signal representation learning from earphone-based sensors. To address data diversity and task specificity limitations of existing approaches, the authors develop a wearable hardware platform (“NeuroBuds”) and collect the DailySense dataset, comprising 50 hours of free-living recordings and 20 hours of labeled, multi-sensory data spanning the five human senses. The proposed Physiology-informed Multi-band Tokenization (PiMT) method decomposes ExG signals into 12 pre-defined, physiologically meaningful sub-bands, enabling generalizable tokenization. The model is pre-trained using self-supervised multi-task reconstruction and then fine-tuned for various downstream tasks. Comprehensive experiments benchmark performance against standard baselines across both the new DailySense dataset and four public ExG datasets, with results indicating significant improvements in generalization and accuracy.

**Strengths:**

1. Ambitious Data Collection and New Benchmark: The DailySense dataset, comprising the largest known free-living ExG recordings across diverse human activities and all five senses, marks a significant step toward real-world applicability for physiological sensing. The use of an earphone-based device (NeuroBuds) demonstrates strong engineering innovation, promising unobtrusive and scalable physiological monitoring.

2. Principled Tokenization Approach: The PiMT framework’s multi-band tokenization sits on a solid physiological basis, decomposing signals into 12 canonical sub-bands. This overcomes the usual artificial rigidity of task-specific band choices and supports more generic, transferable representations. The explicit mapping of ExG modalities to sub-bands is clearly visualized and justified in both Figure 1 and the accompanying methodology.

3. Thorough Experimental Validation: The paper conducts comprehensive comparisons against traditional and state-of-the-art baselines (SVM, DeepConvNet, EEGNet, PatchTST, EEGConformer, Bidirectional-Mamba), with both within-dataset and cross-benchmark evaluations. Ablations and analyses deepen understanding of where the gains come from and how PiMT generalizes.

4. Foundation Model Potential: By combining self-supervised pre-training on free-living data with a flexible multi-band tokenization scheme, PiMT takes a meaningful step toward a foundation model for ExG. The method is not tied to any specific task or sensor configuration, and its performance gains across diverse tasks and datasets suggest that it learns robust, general-purpose representations. This is a notable advance in a field often dominated by narrow, task-specific models.

**Weaknesses:**

1. Limited Generalization to Unseen Subjects and Modest Cohort Size: While the participant count (N=22) is comparable to prior lab-based ExG studies, it remains insufficient to support strong claims of robust population-level generalization. This limitation is clearly evidenced by the significant performance drop in the cross-subject setting (Table 7), where the average F1-score falls to ~58%. Although the authors acknowledge this challenge and provide Leave-One-Subject-Out (LOSO) results (Figure 7), the sharp decline underscores that user-independent modeling remains a substantial hurdle. The current work is a robust proof-of-concept rather than a fully generalizable solution. Future work would benefit from larger-scale recruitment to better address subject variability.

2. Ambiguity in Filter Bank Implementation and Saliency Analysis: The physiological basis for the 12 frequency bands is well-motivated, but the practical implementation lacks critical details, which may hinder reproducibility. The manuscript does not specify key filter parameters (e.g., filter type, order, transition bandwidth, or handling of overlapping bands like EMG-Low and EEG-Beta/Gamma). These details are crucial for replicating the tokenization process.

3. Lack of Statistical Significance Testing: The reporting of mean performance with standard deviations is standard practice, but it is not sufficient to firmly establish the superiority of a proposed method over multiple strong baselines. The absence of statistical significance tests makes it difficult to assess whether the observed improvements (e.g., the 4% F1-score gain in Table 1) are statistically reliable, especially given the variability inherent in physiological data. Incorporating such tests would greatly strengthen the quantitative claims.

4. Insufficient Validation of NeuroBuds Signal Quality Against Gold Standards: A key selling point of the work is the novel NeuroBuds hardware. However, the validation of signal quality is primarily indirect, relying on downstream task performance. To fully establish the device's credibility for research use, a more direct, quantitative comparison with a clinical-grade or research-standard ExG system (even on a small subset of participants) would be highly valuable.

**Questions:**

See Weaknesses.

**Details Of Ethics Concerns:**

No ethics concerns are apparent. The study protocol for data collection is IRB-approved, and all public datasets are used appropriately.

---

> ### Author Response · Authors · 2025-11-23
> **Response to Reviewer ejoF (1/2)**
>
> Dear Reviewer ejoF,
>
> We are grateful for your insightful and constructive comments. We are encouraged by the reviewers’ recognition of our **solid prototype and free-living dataset**, which together enable scalable ExG sensing in the wild (highlighted by mTe5, ejoF, ZFuK). We are also grateful for the positive assessment of our **PiMT and pre-training**, which reviewers found to be principled, novel, and broadly effective across tasks (ejoF, UWPo, ZFuK). Finally, we are heartened by the reviewers’ acknowledgement of our **comprehensive experiments**, including evaluations on both our new dataset and four public benchmarks, as well as the **ablation studies** that clarify the contribution of each component (mTe5, ejoF, UWPo, ZFuK).
>
> We have carefully addressed your comments below.
>
> ---
>
> **[W1] Limited Generalization and Modest Cohort Size**
>
> Thank you for your helpful comments. We agree that achieving strong population-level generalization remains a fundamental challenge in EEG/ExG modeling. As you noted the performance drop in the cross-subject setting reflects an inherent difficulty of user-independent ExG modeling, and our goal in this goal is not to claim that this challenge is fully solved. Rather, our primary contribution focuses on *task-level generalization* across diverse downstream tasks spanning all five human senses—an important step toward enabling broad ExG applications.
>
> We also acknowledge that our cohort size while comparable to prior lab-based ExG studies, larger and more heterogeneous populations will be essential for strengthening population-level generalization. A key advantage of our work is the low-cost, easy-to-wear earphone prototype, which substantially reduces the barrier to recruiting larger and more diverse cohorts compared with traditional bulky EEG systems. As part of future work, we plan to leverage this scalable prototype to build significantly larger datasets that will directly address user-independent generalization at scale.
>
> ---
>
> **[W2] Filter Bank Implementation Details**
>
> In response to the details of filter process, each channel’s time-domain signal is processed through the following steps:
>
> - *Second-order IIR notch filters (Q=30)* are applied at both 50Hz and 60Hz, with zero-phase forward-backward filtering.
> - The filtered signals are *duplicated into 12 parallel branches*, each corresponding to a physiology band (e.g., EEG-beta, EMG-low).
> - Each branch applies *first-order Butterworth low-pass and high-pass filters with zero-phase filtering*, with its corresponding frequency range.
>
> Since each band is obtained by filtering an independent copy of the notch-filtered data, the method does not involve cascaded filtering; overlapping bands do not interfere. Each band acts as an isolated physiological view of the same signal, and results in 12 tokens. Our saliency analysis illustrates how different tasks rely on different physiological views.
>
> We will clarify the details and update Section 4 in the revised manuscript to ensure reproducibility.

---

> ### Author Response · Authors · 2025-11-23
> **Response to Reviewer ejoF (2/2)**
>
> **[W3] Statistical Significance Testing**
>
> To assess whether our method provides a statistically significant gain over the baselines, we conduct paired Wilcoxon signed-rank tests comparing PiMT against each baseline across all tasks. Each task contains 6–9 participants, and for every participant we train three models with different random seeds. All models share the same train–test split, ensuring that each participant–seed pair forms a fair matched comparison. Statistical tests are then applied over these paired samples.
>
> | Method | Video | Audio | Taste | Touch | Smell | Avg. | Gaze |
> |-|-|-|-|-|-|-|-|
> | **Without pre-training** | | | | | | | |
> | SVM | 4.8e-7 (*** ) | 4.8e-7 (*** ) | 7.6e-6 (*** ) | 7.5e-9 (*** ) | 4.8e-7 (*** ) | 9.6e-20 (*** ) | 4.4e-5 (*** ) |
> | EEGNet | 1.6e-4 (*** ) | 1.2e-4 (*** ) | 1.2e-2 (* ) | 1.5e-8 (*** ) | 1.6e-3 (** ) | 8.3e-15 (*** ) | 1.0e-3 (** ) |
> | DeepConvNet | 9.8e-5 (*** ) | 4.8e-7 (*** ) | 1.6e-4 (*** ) | 1.4e-7 (*** ) | 4.8e-6 (*** ) | 4.5e-18 (*** ) | 1.0e-7 (*** ) |
> | TST | 1.2e-4 (*** ) | 1.2e-5 (*** ) | 1.9e-2 (* ) | 4.1e-7 (*** ) | 6.5e-5 (*** ) | 1.2e-15 (*** ) | 8.0e-4 (*** ) |
> | PatchTST | 4.3e-4 (*** ) | 1.1e-4 (*** ) | 3.0e-2 (* ) | 1.5e-8 (*** ) | 1.7e-3 (** ) | 2.7e-14 (*** ) | 2.1e-3 (** ) |
> | EEGConformer | 9.8e-5 (*** ) | 5.3e-5 (*** ) | 5.2e-4 (*** ) | 1.4e-7 (*** ) | 1.2e-3 (** ) | 3.3e-16 (*** ) | 1.4e-3 (** ) |
> | Bidirectional-Mamba | 6.1e-3 (** ) | 1.8e-2 (* ) | 1.6e-2 (* ) | 1.1e-3 (** ) | 2.3e-2 (* ) | 1.0e-7 (*** ) | 1.2e-7 (*** ) |
> | PiMT (Ours) | -- | -- | -- | -- | -- | -- | -- |
> | **With pre-training** | | | | | | | |
> | PatchTST | 1.2e-4 (*** ) | 9.5e-7 (*** ) | 1.3e-4 (*** ) | 7.5e-8 (*** ) | 9.5e-7 (*** ) | 1.9e-18 (*** ) | 3.3e-3 (** ) |
> | PiMT (Ours) | -- | -- | -- | -- | -- | -- | -- |
>
> As shown in table above, PiMT achieves statistically significant improvements over nearly all baselines and modalities with extremely small $p$-values  (p < 0.001), demonstrating that the gains are consistent and robust across participants and seed-level variations.  These results, along with a clearer description of the evaluation protocol, have been added to the Appendix M in the revised manuscript.
>
> ---
>
> **[W4] Validation of Signal Quality**
>
> In addition to validating signal utility through downstream tasks, we conducted a direct comparison between our earphone-based NeuroBuds system and a research-grade OpenBCI device to more explicitly assess signal quality. To ensure a fair comparison, we placed NeuroBuds and OpenBCI electrodes in matched, neighboring positions and collected approximately one hour of synchronized data from two participants during an eye-movement tracking experiment. Following the evaluation protocol in [r1], we computed the Pearson correlation between the corresponding NeuroBuds and OpenBCI ExG channels. The average cross-system correlation was 0.71 (p < 0.001), indicating that NeuroBuds captures highly consistent ExG activity relative to a research-grade device.
>
> We also qualitatively examined raw synchronized signals. As shown in Figure 9 in Appendix D, the NeuroBuds and OpenBCI waveforms exhibit closely aligned temporal patterns, including similar shapes, amplitudes, and drift trends. Together with the correlation analysis, these observations support that NeuroBuds provides ExG signals that closely match those from established research-grade equipment. We have updated Appendix D in the revised manuscript to include this experiment and the accompanying discussion.
>
> [r1] Frey, Jérémy. "Comparison of an open-hardware electroencephalography amplifier with medical grade device in brain-computer interface applications." PhyCS-ICPCS 2016.

---

### Official Review · Reviewer_mTe5 · 2025-11-01

**Soundness:** 3
**Presentation:** 3
**Contribution:** 4
**Rating:** 4
**Confidence:** 4

**Summary:**

This work built an earphone-based prototype for ExG data collection. The paper built a dataset consisting of 50 hours of free-living data, and 20 hours of task-specific data based on this new prototype. The paper further proposed a machine learning method called physiology-informed multi-band tokenization (PiMT) that learn representations from the ExG signals. The method is experimentally validated on both the collected dataset, and also four public ExG benchmarks.

**Strengths:**

- Solid prototype building and dataset collection efforts. The dataset is gonna be highly valuable to the community if made public, especially considering its multimodality nature and the study design that covers a wide range of tasks of interest.
- Solid experimental efforts. The method is applied on a wide range of tasks, including both private datasets and public datasets, and showed good performance.
- Very interesting saliency analysis, showing both how different frequency components are being effective differently on different tasks, and also shows how the multimodal ExG dataset is being effective such that it allows researchers to analyze such frequency components.
- Interesting experiments in section 5.5, showing increasing pre-training data at scale can improve performance.

**Weaknesses:**

1. It is unclear if the self-collected dataset is superior comparing to existing non free-living datasets. Specifically:
- The paper lacks experiments showing how the free-living dataset compares to existing larger-scale pre-training datasets, for example, the multimodal sleep datasets that contains thousands of hours of data (You snooze you win challenge, or TU datasets, as used in [1, 2, 3]). The paper hypothesize the potential benefits of collecting free-living ExG data, but the experimental results do not demonstrate as such. To demonstrate the self-collected datasets are effective, the authors should consider comparing the pre-training benefits based on the new dataset, and compare against the pre-training benefits given by using (1) 50 hours of existing datasets; (2) >>50 hours of existing datasets, and compare performance differences.

[1] Chien, H. Y. S., Goh, H., Sandino, C. M., & Cheng, J. Y. (2022). Maeeg: Masked auto-encoder for eeg representation learning. arXiv preprint arXiv:2211.02625.

[2] Liu, Ran, Ellen L. Zippi, Hadi Pouransari, Chris Sandino, Jingping Nie, Hanlin Goh, Erdrin Azemi, and Ali Moin. "Frequency-aware masked autoencoders for multimodal pretraining on biosignals." arXiv preprint arXiv:2309.05927 (2023).

[3] Fang, Ching, Christopher Sandino, Behrooz Mahasseni, Juri Minxha, Hadi Pouransari, Erdrin Azemi, Ali Moin, and Ellen Zippi. "Promoting cross-modal representations to improve multimodal foundation models for physiological signals." arXiv preprint arXiv:2410.16424 (2024).

2. The technical details of the proposed method are unclear, and the reported numbers are way off the range of existing numbers, leading to doubts and questions about their experimental settings. For example, in papers like [4], they report around 40% balanced accuracy on SEED-V dataset and around 54% balanced accuracy on BCI competition, which is the state-of-the-art performance that is commonly reported. Yet in this paper, there is 82% accuracy on SEED dataset (which one?) and 69% accuracy on BCI competition, which leads to concerns - have the authors used the common metrics like balanced accuracies? Also, the performance reported on SleepEDF seems to be quite low, see [2, 5], which both reported 83%-85% accuracy on SleepEDF. The experimental results inconsistency makes it hard to evaluate if the machine learning method is effective.

[4] Wang, Jiquan, Sha Zhao, Zhiling Luo, Yangxuan Zhou, Haiteng Jiang, Shijian Li, Tao Li, and Gang Pan. "Cbramod: A criss-cross brain foundation model for eeg decoding." arXiv preprint arXiv:2412.07236 (2024).

[5] Eldele, Emadeldeen, Mohamed Ragab, Zhenghua Chen, Min Wu, Chee Keong Kwoh, Xiaoli Li, and Cuntai Guan. "Time-series representation learning via temporal and contextual contrasting." arXiv preprint arXiv:2106.14112 (2021).


Overall, the contribution of the prototype and the self-collected dataset is a bit disconnected from the machine learning perspective. I’d suggest the authors to design the experiments differently, to either showcase the effectiveness of the dataset, or the effectiveness of the machine learning approach.

**Questions:**

Have the author considered submitting the dataset to a journal instead?

---

> ### Author Response · Authors · 2025-11-23
> **Response to Reviewer mTe5 (1/3)**
>
> Dear Reviewer mTe5,
>
> We deeply appreciate your constructive reviews. We are encouraged by the reviewers’ recognition of our **solid prototype and free-living dataset**, which together enable scalable ExG sensing in the wild (highlighted by mTe5, ejoF, ZFuK). We are also grateful for the positive assessment of our **PiMT and pre-training**, which reviewers found to be principled, novel, and broadly effective across tasks (ejoF, UWPo, ZFuK). Finally, we are heartened by the reviewers’ acknowledgement of our **comprehensive experiments**, including evaluations on both our new dataset and four public benchmarks, as well as the **ablation studies** that clarify the contribution of each component (mTe5, ejoF, UWPo, ZFuK).
>
> We have carefully addressed your feedback and questions below.
>
> ---
>
> **[W1] Comparison with Larger-scale Pre-training Datasets**
>
> We would like to clarify that our goal is not to claim that our 50-hour free-living dataset is larger than prior datasets, but rather to bridge an important gap in ExG research; existing datasets 1) are lab-controlled, 2) use bulky devices, and 3) mostly focus on a single task. To overcome them, we propose  unobtrusive, earphone-based, multi-modal ExG collected during unconstrained daily life and evaluated across diverse tasks.
>
> Based on your suggestions, we conducted new experiments comparing the pre-training benefits of *DailySense* (50 hrs) against (1) TUAR (98.6 hrs, moderate-scale baseline) [r1] and (2) TUAR combined with a subset of TUSZ [r2] (498.6 hrs, large-scale baseline). Because these datasets use electrode montages that differ from ours, we select electrodes with the closest spatial correspondence to align with our setting—F7, F8, T3, T4, T5, T6, O1, and O2 in the 10–20 system—for pretraining. All models were pre-trained using identical training protocols for fairness.
>
> | Method | Video | Audio | Taste | Touch | Smell | Avg. | Gaze |
> |-|-|-|-|-|-|-|-|
> | w/o pt | 0.858 | 0.885 | 0.790 | 0.807 | 0.753 | 0.819 | 6.11 |
> | TUAR (98.6 hrs) | 0.964 | 0.950 | 0.791 | 0.824 | 0.736 | 0.853 | 5.95 |
> | TUAR + TUSZ (498.6 hrs) | 0.964 | 0.950 | 0.803 | 0.833 | 0.741 | 0.858 | 6.03 |
> | **DailySense (Ours, 50 hrs)** | **0.964** | **0.961** | **0.801** | **0.860** | **0.793** | **0.876** | **6.00** |
>
> As shown in table, pretraining on *DailySense* achieves the strongest average performance across all classification tasks and yields competitive gaze estimation accuracy, despite its substantially smaller size. This demonstrates that the free-living distribution provides more transferable representations than larger but controlled datasets. The results validate that our dataset is effective not because of scale, but because it captures real-world variability absent in laboratory EEG recordings. Detailed experiment setups are in updated Appendix N.
>
> [r1] Hamid, et al. "The temple university artifact corpus: An annotated corpus of eeg artifacts." IEEE SPMB 2020.
>
> [r2] Shah, et al. "The temple university hospital seizure detection corpus." Frontiers in neuroinformatics 2018.

---

> ### Author Response · Authors · 2025-11-23
> **Response to Reviewer mTe5 (2/3)**
>
> **[W2] Clarification on Experimental Results**
>
> We emphasize all experimental setting within our paper were compared under a *fair and consistent setting*. We applied the same data processing and splits across all methods to isolate the gain from PiMT. We also used *macro-averaged F1-score* as our metric for a balanced evaluation under class-imbalanced conditions across datasets.
>
> In response to your question, reported numbers in prior work and in our paper appear differently mainly because datasets mismatch and metric mismatch. Below we clarify the differences and show that our results are consistent and reproducible when aligned under a unified setting:
>
> 1. Datasets mismatch explains numerical differences
>
> - [4] reports results on SEED-V (Liu et al., 2021), whereas we use the original SEED (Duan et al., (2013)), which differs in class structures.
> - [4] uses BCI Competition IV (i.e., BCIC) 2a (Brunner et al., 2008), while we use BCIC 2b (Leeb at al., 2008). These datasets differ in classes and sensor placements.
> - [2] uses SleepEDF for pre-training and evaluates on SleepEOG, a subset with only EOG channels (reporting F1 = 0.66).
>
> 2. Metric mismatch explains confusion
>
> - [5] reports 83% accuracy, but the corresponding F1-score is 0.73, and their setting uses only a single EEG channel. Our setting uses EEGs+EOG+EMG, achieving 0.82 F1-score. The modality difference explains the performance gap.
>
> To clarify, our choice of SEED and BCIC 2b follows EEGConformer [r3], a well-established benchmark in EEG deep learning. Nevertheless, EEGConformer reports 95.3% and 84.6% accuracy (F1-scores were similar when reproduced) on the two datasets, which differ from our initial numbers due to differences in data processing strategies. Following your recommendation, to ensure a fair comparison with the literature, we adopted EEGConformer’s official code for data processing, and reran our baselines and PiMT under a standardized setting.
>
> | Method | SEED | BCI Competition IV 2b |
> |-|-|-|
> | PatchTST | 0.945 | 0.826 |
> | Bidirectional-Mamba | 0.956 | 0.823 |
> | EEGConformer| 0.955 | 0.855 |
> | PiMT | 0.971 | 0.869 |
>
> The table above shows that our results align well with existing literature, and that PiMT consistently improves over all baselines in this standardized configuration. We will revise the manuscript to explicitly highlight datasets details, metric selection and preprocessing setting in Section 5.1.
>
> [r3] Song, et al. "EEG conformer: Convolutional transformer for EEG decoding and visualization." IEEE TNSRE 2022.

---

> ### Author Response · Authors · 2025-11-23
> **Response to Reviewer mTe5 (3/3)**
>
> **[W3-1] Motivation for the Prototype and Self-collected Dataset**
>
> We would like to clarify that the hardware prototype, the free-living dataset and our ML method are tightly connected. Our goal is to learn a general task-agnostic ExG model that can generalize across diverse physiological tasks with a single model. However, ExG models are limited because they are trained on datasets collected in controlled environments using bulky hardware and focus on narrow, single-task settings.  As a result, these datasets lack key characteristics both 1) to build **pre-train generalizable models** and 2) to serve as **realistic benchmarks** for evaluating different tasks in a unified setting.
>
> This gap directly motivated both our prototype design and data collection. The prototype enables ExG sensing in daily life, which existing devices do not support. Using it, we collected a free-living dataset that captures the diversity required to (1) pre-train models under realistic conditions and (2) establish the first benchmark for evaluating ExG models across different tasks. Building on this foundation, PiMT was developed to effectively handle the characteristics of real-world ExG signals that controlled-setting models cannot reliably address. Together we believe the prototype, dataset and ML model form an integrated co-design, where the dataset is not auxiliary but the driving component that motivates and enables our ML approach.
>
> ---
>
> **[W3-2] Effectiveness of the Dataset and ML Approach**
>
> We appreciate your pointer. In response, we conducted additional analyses to separately demonstrate the effectiveness of our free-living dataset and our PiMT architecture.
> - **Effectiveness of datasets**: as shown in mTe5 [W1], we conducted new experiments demonstrating that pretraining on our free-living dataset leads to more robust and transferable representations than pretraining on larger but controlled-setting datasets, validating the effectiveness of DailySense dataset.
> - **Effectiveness of PiMT architecture**: as shown in mTe5 [W2], PiMT consistently outperforms baselines across both DailySense and public benchmarks under unified evaluation, confirming the effectiveness of the ML design independent of the dataset. Further, we performed analysis to isolate the effect of PiMT (UWPo [Q1]) and compared PiMT against alternative tokenizations (ZFuK [1-1]), which showcase PiMT provides clear improvements beyond backbone and standard tokenization methods.
>
> With the updated results, we will highlight the effectiveness of our dataset and ML approach more clearly in the revised manuscript.
>
> ---
>
> **[Q1] Future Direction**
>
> We appreciate your suggestion. We believe that with our low-cost and easy-to-wear earphone prototype, the platform has a strong potential to scale toward larger and more diverse ExG datasets collected in free-living conditions, enabling broader applications. We will keep working along this line and are actively exploring these possibilities.

---

### Author Response · Authors · 2025-11-23
**Summary of the Rebuttal**

We thank all reviewers for their constructive feedback. We are encouraged by the reviewers’ recognition of our **prototype and free-living dataset** (mTe5, ejoF, ZFuK), the novelty and effectiveness of our **PiMT and pre-training** (ejoF, UWPo, ZFuK), and the strength of our **comprehensive experiments** across both DailySense and public benchmarks, along with the **ablation studies** validating each component (mTe5, ejoF, UWPo, ZFuK).

We carefully addressed all concerns raised during the review process. In particular, we conducted substantial new experiments and added key clarifications, including:

1. **Validated the effectiveness of our free-living dataset** by comparing DailySense with larger lab pre-training datasets, where it yields the strongest transfer performance (mTe5 W1);

2. **Demonstrated the effectiveness of PiMT** by showing clear gains beyond backbone variants and standard tokenizations (mTe5 W3-2; UWPo Q1; ZFuK W1-1);

3. **Validated robustness** through statistical significance tests and cross-device signal-quality experiments (ejoF W3, W4);

4. **Strengthened representation analyses** through frozen-encoder and linear-probing evaluations (UWPo Q3; ZFuK W1-2);

5. **Clarified implementation details**, including consistency with with prior work, PiMT’s multi-band filter-bank design, and minimal-preprocessing rationale (mTe5 W2, ejoF W2; UWPo W2);

6. **Discussed generalization scope** and highlighted the scalability of our ear-ExG prototype for future larger-cohort studies (ejoF W1; UWPo W1; UWPo Q2; mTe5 Q1).

We hope the revisions fully address the raised concerns, and we welcome any further questions from ACs or reviewers.

---

### Author Response · Authors · 2025-11-30
**Letter to Area Chairs (1/2)**

Dear Area Chairs,

We would like to express our sincere appreciation for your huge effort during this challenging situation. To support your understanding of our revisions, we have provided a consolidated summary of the reviews and our corresponding responses.

First, we are very grateful for the overall support from the reviewers. We are pleased that they highlighted several key strengths of our work:

1. **Solid hardware prototype and free-living DailySense dataset**, enabling scalable ExG sensing in the real-world condition (mTe5, ejoF, ZFuK).
2. **Principled PiMT method and Pre-training**, recognized as novel and broadly effective across diverse tasks (ejoF, UWPo, ZFuK).
3. **Comprehensive experiments**, including thorough evaluations on both our DailySense dataset and four public benchmarks (mTe5, ejoF, UWPo, ZFuK).
4. **Solid ablation studies and interesting saliency analysis**, clarifying the specific contribution of each component (mTe5, ejoF, UWPo, ZFuK).
5. **Foundation model potential**, marking a meaningful step toward general-purpose ExG models (ejoF, UWPo).

We also greatly appreciate the constructive comments raised by the reviewers, which helped us significantly strengthen our work. We have carefully addressed each point, and below we provide a point-to-point summary.

---

**Reviewer UWPo**

*Latest discussion status: the reviewer replied that **all questions were fully addressed**; **maintained score of 8**.*

W1: Generality of PiMT to different hardware configurations.

> We clarified that PiMT is designed to *be flexible across hardware and does not assume physiological separation at specific electrodes*. It uses a broad physiology-inspired band set, allowing the model to learn *whichever features are informative for each task and each electrode*.

W2: Baseline models may not perform optimally under minimal preprocessing.

> We clarified that minimal preprocessing is used to maintain *a fair, task-agnostic comparison*. We explained that additional processing would not generalize across tasks. We evaluated *CWT- and STFT-based variants* and showed that PiMT still outperforms these alternatives.

Q1: What is the performance without PiMT but with pre-training, and does it work with another architecture?

> We added *experiments isolating PiMT* and showed that pre-training alone improves performance. We also *evaluated a Transformer backbone* and confirmed that the gains remain consistent.

Q2: Can PiMT help with cross-subject generalization?

> We clarified that *PiMT shows promising representation-level transfer to unseen users*, but full cross-subject generalization remains an open challenge. PiMT may help capture subject-invariant components, but confirming this will require larger datasets.

Q3: Performance when the pre-trained encoder is frozen.

> We conducted *frozen-encoder experiments* showing that PiMT’s pre-trained representations transfer well across tasks.

---

**Reviewer ZFuK**

*Latest discussion status: the reviewer replied that **all concerns were fully addressed**; **score raised 6 → 8**.*

W1-1: Request for experiments using alternative backbones and tokenization approaches.

> We conducted experiments showing that (1) *PiMT improves performance even when using a Transformer backbone*, and (2) *PiMT outperforms alternative tokenization (STFT- and CWT-based) approaches* on DailySense tasks.

W1-2: Request for comparison of learned representation with other recent open-sourced pre-trained models.

> We ran *linear probing experiments* comparing PiMT’s pre-trained representations against *recent channel-agnostic models (CBraMod [r1], NormWear [r2])*. PiMT achieved the strongest transfer performance, confirming the quality of the learned features.

W2-1: The size of the data is a bit limited.

> We clarified that *DailySense (70 hrs total) exceeds or matches existing in-lab ExG datasets*, and our unobtrusive prototype enables natural long-duration recordings, supporting future large-scale expansion.

W2-2: The downstream tasks of DailySense are relatively straightforward.

> We explained that *the DailySense tasks follow established EEG protocols [r3, r4, r5]*, and we further validated PiMT on more *complex public benchmarks* (e.g., 3-class SEED, 5-class Sleep-EDF), where PiMT performs strongly.

W2-3: A data sheet would improve presentation.

> We added *a data sheet summarizing DailySense and public datasets* for clarity.

---

> ### Author Response · Authors · 2025-11-30
> **Letter to Area Chairs (2/2)**
>
> **Reviewer ejoF**
>
> *Latest discussion status: no response from the reviewer*
>
> W1: Limited generalization and modest cohort size.
>
> > We clarified that cross-subject generalization remains an open challenge and *our contribution focuses on task-level generalization*. We emphasized that our prototype enables *scalable future data collection*, and we plan to expand to larger cohorts.
>
> W2: Ambiguity in filter bank implementation.
>
> > We added *full implementation details for the filter bank*, ensuring reproducibility.
>
> W3: Lack of statistical significance testing.
>
> > We added *paired Wilcoxon signed-rank tests comparing PiMT against all baselines*. We reported that PiMT achieves statistically significant improvements (p < 0.001).
>
> W4: Insufficient validation of signal quality against gold standards.
>
> > We conducted *a direct comparison between NeuroBuds and a research-grade OpenBCI device using synchronized recordings*. Using the protocol in [r6], we observed a cross-signal correlation of 0.71 (p < 0.001), along with strong alignment of waveform patterns.
>
> ---
>
> **Reviewer mTe5**
>
> *Latest discussion status: no response from the reviewer*
>
> W1: It is unclear whether DailySense is superior to existing non-free-living datasets.
>
> > We conducted new experiments *comparing pre-training on DailySense against two other datasets*, (1) TUAR [r7] (98.6 hrs) and (2) TUAR + a TUSZ [r8] subset (498.6 hrs), and found that DailySense yields stronger overall performance.
>
> W2: The reported experimental results appear inconsistent with those in prior literature [r1, r9, r10].
>
> > We found that *the referenced works [r1, r9, r10] use different datasets* with similar names but differ in class and task or different metrics, which explains the numerical mismatch. To ensure clarity, we *reproduced the exact experimental setup from an established baseline [r11]* and confirmed that our method consistently outperforms the baselines.
>
> W3-1: Concern about disconnect between prototype/dataset and ML contribution.
>
> > We clarified that *the prototype and the dataset are key enablers for our task-agnostic PiMT framework*, providing the real-world signals and task diversity needed to make the ML approach feasible.
>
> W3-2: Paper should emphasize effectiveness of the dataset and ML approach.
>
> > We highlighted the effectiveness of both DailySense and PiMT through the *new experiments in W1 and W2*, along with *additional ablations isolating the contribution of PiMT* (UWPo Q1, ZFuK W1-1).
>
> ---
>
> We have thoroughly addressed all concerns raised by the reviewers, and our refined manuscript will incorporate all the points discussed in our rebuttal. We believe that these enhancements will further clarify the significance of our work. Once again, we sincerely appreciate your valuable contributions.
>
> Warm regards,
>
> Authors
>
> > References
> >
> > [r1] Wang, et al. "Cbramod: A criss-cross brain foundation model for eeg decoding." ICLR 2025.
> >
> > [r2] Luo, et al. "Toward Foundation Model for Multivariate Wearable Sensing of Physiological Signals." arXiv 2024.
> >
> > [r3] Amini, et al. "Surface roughness classification in dynamic touch using EEG signals." ICEE 2022.
> >
> > [r4] Eldeeb, et al. "EEG-based trial-by-trial texture classification during active touch." Scientific reports 2020.
> >
> > [r5] Libert, et al. "Predicting premature video skipping and viewer interest from EEG recordings." Entropy 2019.
> >
> > [r6] Frey, Jérémy. "Comparison of an open-hardware electroencephalography amplifier with medical grade device in brain-computer interface applications." PhyCS-ICPCS 2016.
> >
> > [r7] Hamid, et al. "The temple university artifact corpus: An annotated corpus of eeg artifacts." IEEE SPMB 2020.
> >
> > [r8] Shah, et al. "The temple university hospital seizure detection corpus." Frontiers in neuroinformatics 2018.
> >
> > [r9] Liu, et al. "Frequency-aware masked autoencoders for multimodal pretraining on biosignals." ICLR 2024 workshop.
> >
> > [r10] Eldele, et al. "Time-series representation learning via temporal and contextual contrasting." IJCAI 2021.
> >
> > [r11] Song, et al. "EEG conformer: Convolutional transformer for EEG decoding and visualization." IEEE TNSRE 2022.

---

### Meta-Review · Area_Chair_qAG7 · 2026-01-02

**Summary:**

In sum, I support accepting this paper as a poster to ICLR 2026.

Here are the main concerns from the reviewers:

Reviewer UWPo:

1. What is the performance without PiMT but with pre-training, and does it work with another architecture? The authors have provided additional experiments showing that pre-training improves performance across both architectures without PiMT, but the gains from PiMT are substantially larger.
2. Cross-subject generalization: The authors argue that the downstream performance of pre-training PiMT excluding the target user’s free-living data is comparable to full pre-training with target user’s data, but cross-subject testing is challenging.
3. Frozen encoder without task-specific tuning: the authors conducted frozen-encoder experiments showing that PiMT’s pre-trained representations transfer well across tasks.

Reviewer ZFuK:
1. Request for experiments using alternative backbones and tokenization approaches: the authors conducted experiments showing that (1) PiMT improves performance even when using a Transformer backbone, and (2) PiMT outperforms alternative tokenization (STFT- and CWT-based) approaches on DailySense tasks.
2. Request for comparison of learned representation with other recent open-sourced pre-trained models. The authors ran linear probing experiments comparing PiMT’s pre-trained representations against recent channel-agnostic models (CBraMod, NormWear). PiMT achieved the strongest transfer performance.

Reviewer ejoF:
1. Limited generalization (cross-subject generalization) and modest cohort size. The reviewer points out that a significant performance drop (around 58%) in the cross-subject setting (Table 7). The authors argue that the focus of the paper is more on task generalization.
2. Statistical significance: The authors have conducted Wilcoxon signed-rank tests.

Reviewer mTe5:
1. It is unclear whether DailySense is superior to existing non-free-living datasets. The authors conducted new experiments comparing pre-training on DailySense against two other datasets, (1) TUAR and (2) TUAR + a TUSZ subset (498.6 hrs), and found that DailySense yields stronger overall performance.
2. Inconsistent results with prior works. The authors reproduced the exact experimental setup from an established baseline and confirmed that their method consistently outperforms the baselines.

Overall, this paper combines elements of both a benchmark paper and a methodological contribution. However, the impact in each direction is somewhat limited: the dataset remains relatively small, and the modeling component introduces only modest novelty. That said, the paper provides a well-executed example of how dataset creation and representation learning (Physiology-informed Multi-band Tokenization) can be meaningfully integrated. For this reason, I still recommend acceptance, but as a poster paper, not above.

**Reviewer Concerns:**

For Reviewer UWPo and mTe5, I think major issues are addressed. For Reviewer ZFuK and Reviewer ejoF, I think the question regarding cross-subject generalization is not sufficiently addressed.

**Reviewer Scores:**

Reviewer UWPo replied that all questions were fully addressed and maintained score of 8, which I agree.
Reviewer ZFuK raised the score from 6 to 8, which I agree.
Reviewer ejoF has not replied, but since the authors have not answered the cross-subject generation question well, I think the reviewer may maintain their score as 6.
Reviewer mTe5 has not replied, but I project there will be a score increase to 6.

---

### Decision · Program_Chairs · 2026-01-26

Accept (Poster)